



# Improving Surface Melt Estimation over Antarctica Using Deep Learning: A Proof-of-Concept over the Larsen Ice Shelf

Zhongyang Hu[1], Peter Kuipers Munneke[1], Stef Lhermitte[2], Maaike Izeboud[2], and Michiel van den Broeke[1]

[1]Institute for Marine and Atmospheric research Utrecht, Utrecht University, Utrecht, The Netherlands
[2]Department of Geoscience & Remote Sensing, Delft University of Technology, Delft, The Netherlands

**Correspondence:** Zhongyang Hu (z.hu@uu.nl)

**Abstract.** Accurately estimating surface melt volume of the Antarctic Ice Sheet is challenging, and has hitherto relied on climate modelling, or on observations from satellite remote sensing. Each of these methods has its limitations, especially in regions with high surface melt. This study aims to demonstrate the potential of improving surface melt simulations by deploying a deep learning model. A deep-learning-based framework has been developed to correct surface melt from the

regional atmospheric climate model version 2.3p2 (RACMO2), using meteorological observations from automatic weather stations (AWSs), and surface albedo from satellite imagery. The framework includes three steps: (1) training a deep multilayer perceptron (MLP) model using AWS observations; (2) correcting moderate resolution imaging spectroradiometer (MODIS) albedo observations, and (3) using these two to correct the RACMO2 surface melt simulations. Using observations from three AWSs at the Larsen B and C Ice Shelves, Antarctica, cross-validation shows a high accuracy (root mean square error = 0.95

mm w.e. per day, mean absolute error = 0.42 mm w.e. per day, and coefficient of determination = 0.95). Moreover, the deep MLP model outperforms conventional machine learning models (e.g., random forest regression, XGBoost) and a shallow MLP model. When applying the trained deep MLP model over the entire Larsen Ice Shelf, the resulting, corrected RACMO2 surface melt shows a better correlation with the AWS observations for two out of three AWSs. However, for one location (AWS 18) the deep MLP model does not show improved agreement with AWS observations, likely due to the heterogeneous drivers of

melt within the corresponding coarse resolution model pixels. Our study demonstrates the opportunity to improve surface melt simulations using deep learning combined with satellite albedo observations. On the other hand, more work is required to refine the method, especially for complicated and heterogeneous terrains.

## 1 Introduction

The Antarctic ice sheet (AIS) is an important indicator for climate change. Current AIS mass loss is estimated at $155 \pm 19$

$\mathrm{Gt\,yr^{-1}}$ ($0.43 \pm 0.05 \mathrm{\,mm\,yr^{-1}}$ of eustatic sea level rise) between 2006 and 2015, and accelerating (Pörtner et al., 2019). At present, mass loss is driven mainly by ice shelf weakening due to basal melt (The IMBIE team et al., 2018) or damage processes (Lhermitte et al., 2020) or hydrofracturing due to surface melt (The IMBIE team et al., 2018). In the coming centuries, surface melt is projected to increase strongly over Antarctica (Trusel et al., 2015), increasing the incidence of surface melt-related



instability of ice shelves. In this context, accurate information about surface melt can directly enhance our understanding of
the AIS evolution and its contribution to sea level rise.

Despite the importance of melt volumes estimates, it is not straightforward to derive them accurately from satellite observations or (regional) climate models. Satellite estimates rely, for example, on proxies of melt presence from changes in albedo
(Steffen et al., 1993; Pirazzini, 2004), brightness temperature (Zheng et al., 2020, 2019) or backscatter (Trusel et al., 2013,
2012) to empirically estimate melt flux or melt volumes. However, these satellite methods face difficulties as they often require
locally adapted thresholds (e.g. thresholds in Trusel et al., 2013), or potentially underestimate the melt fluxes over, for example,
blue ice areas (Arthur et al., 2020; Lenaerts et al., 2017) where the contrast between melt and no-melt is less clear.

(Regional) climate models, on the other hand, face difficulties to accurately estimate surface melt over areas with low sur-
face albedo. Often, features of strong surface melt (ponds, blue ice, lakes) are smaller than the model resolution, and processes
that lead to their appearance and dynamics are usually not well represented (Lenaerts et al., 2017; Kingslake et al., 2017).
Optical remote sensing does provide high-quality albedo observations at different spatiotemporal resolutions. It is hence a
competent additional source of data to the albedo simulation from a physics-based climate model. Therefore, we propose a
deep learning method that uses the albedo observations from remote sensing to correct for the shortcomings of climate models.
To date, deep learning has been widely applied in Earth system science to analyze and correct mismatches between model
simulations and observations (Reichstein et al., 2019) as they execute much faster than physics-based models.

Our study aims to develop a novel framework correcting the model-observation mismatch of surface melt in the AIS with
a deep learning model, which utilizes inputs from the physics-based model, the regional atmospheric climate model version
2.3p2 (RACMO2), and remote sensing albedo observations. To achieve this, our study has two primary objectives: (1) develop
a deep learning model to correct the simulations of surface melt from RACMO2 based on automatic weather station (AWS)
observations, as well as (2) apply and evaluate the performance of the developed model in correcting the surface melt simulations from RACMO2. To prove the concept of this framework, we apply it to RACMO2 model simulations over the Larsen Ice
Shelf between 2009 and 2016, using meteorological parameters from RACMO2 and remote sensing observations of surface
albedo. In section 2, we introduce the investigated area and specify all data sets. The architecture of the method and its details
are described in section 3. Sections 4 and 5 present and discuss the results, followed by a summary in section 6.

## 2   Study area and data

### 2.1   Study area

We apply the deep learning framework to the Larsen Ice Shelf, situated to the east of the Antarctic Peninsula. According to
existing estimates, this area produces about 50 to 60 % of all surface meltwater in Antarctica (Kuipers Munneke et al., 2012a;
Trusel et al., 2013). On average, surface melt occurs on 25 days per year in the southern part of Larsen C Ice Shelf, to over



days per year in the western and northern part of the region (Luckman et al., 2014). The Larsen Ice shelf is an ideal test location for developing a framework to improve surface melt estimates, because (1) there is abundant melt; (2) high-quality multi-year AWS data suitable for melt calculations (i.e. including the surface radiation budget) are scarce in Antarctica (Jakobs
et al., 2020) and (3) a previous comparison between RACMO2 albedo and scatterometry revealed that RACMO2 melt can be both higher and lower than observations (Trusel et al., 2013). Thus, the versatility of the method is tested both for enhancing and reducing surface melt.

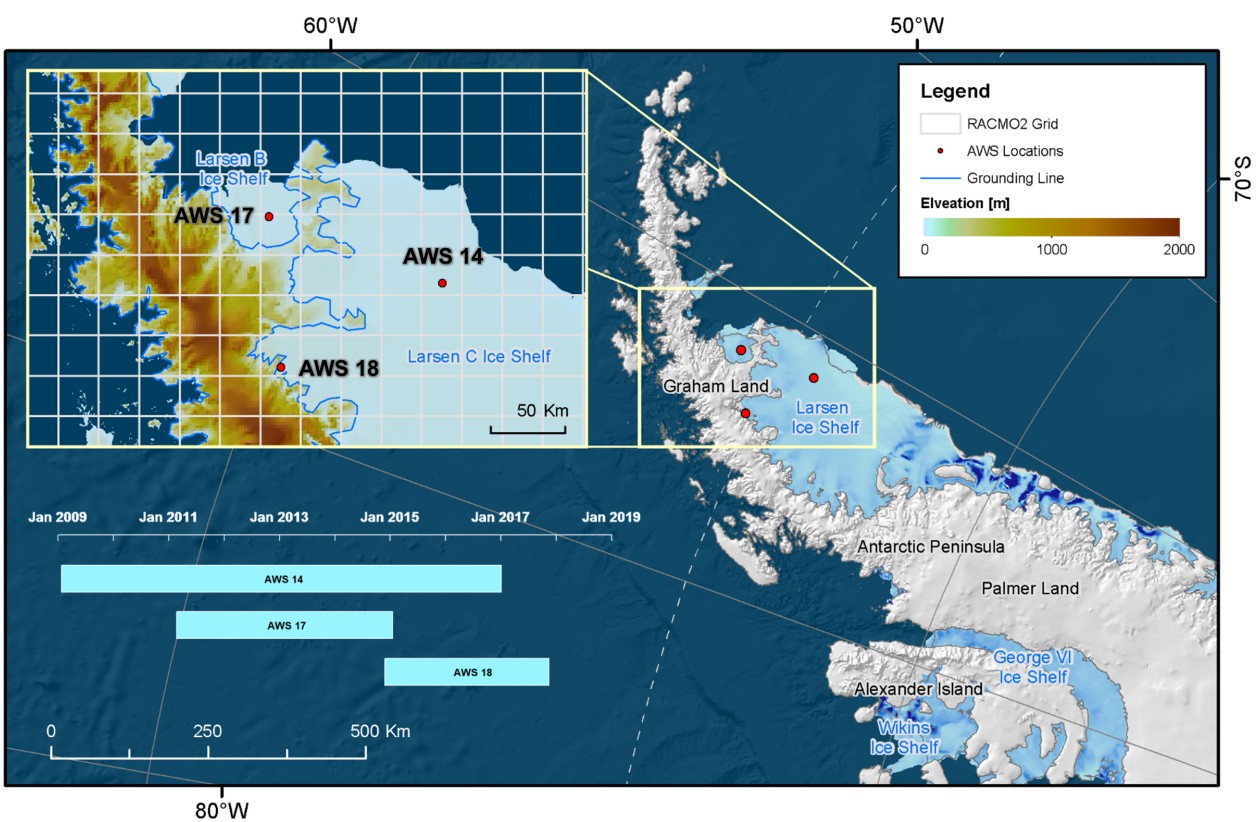

**Figure 1.** Overview of the study area, the Larsen Ice Shelf, and details about the geolocations and operation periods of the deployed automatic weather station (AWS) observations. The elevation information are derived from the ETOPO1 1 arc-minute global relief model (doi:10.7289/V5C8276M), and the grounding line information are derived from Bindschadler et al. (2011). The basemaps are the moderate resolution imaging spectroradiometer (MODIS) mosaic and the Antarctic ice-shelf buttressing (Fürst et al., 2016).

The Antarctic Peninsula is the mildest region of Antarctica, as it is protruding farther north than other regions, into the Southern Ocean. in In the western part of the Antarctic Peninsula the atmospheric circulation is northwest-southeast, leading to mild
conditions, no ice shelves and little sea ice. Conversely, in the eastern Antarctic Peninsula, the circulation is south-north, resulting in colder conditions, extensive ice shelves and year-round sea ice cover. Therefore, ice shelves extend from the Antarctic Peninsula almost only at its eastern coast. Under specific conditions, westerly atmospheric flow leads to warm and



dry winds, known as *föhn*, descending from the eastern mountain flanks onto the ice shelves (Elvidge and Renfrew, 2016; Datta et al., 2019). These föhn winds are known to generate strong surface melt in the inlets on the ice shelves just downslope

from the mountains. On average, the annual melt exceeds $400 \, \mathrm{mm \, w.e.}$ in these inlets, distributed over about 100 melt days. But also further east on the Antarctic Peninsula ice shelves, surface melt rates are high compared to most other ice shelves in Antarctica, at 200 to $300 \, \mathrm{mm \, w.e.}$ per year.

### 2.2   Data

#### 2.2.1   Satellite observations

MODIS aboard the Terra (launched in 1999) and Aqua (launched in 2002) satellites provides continuous observation of the Earth's surface. For various disciplines, there are different standard MODIS data products for global change studies. Among these products, we deployed the bi-hemispherical reflectance (i.e. white-sky albedo) for shortwave broadband from the MCD43A3 albedo product (Schaaf and Wang, 2015) archived in the *Google Earth Engine* (GEE, Gorelick et al., 2017) as the albedo input. Furthermore, to obtain observational information about cloud coverage at AWS locations, cloud classifi-

cations are taken from the MOD09GA daily surface reflectance product (i.e. 'MODIS/Terra Surface Reflectance Daily L2G Global 1 km and 500 m SIN Grid', Vermote and Wolfe, 2015), also archived in GEE. To demonstrate the spatiotemporal melt pattern in the study area, we derived the backscattering coefficient drops as an indicator surface melt following Luckman et al. (2014) and Datta et al. (2019) from some representative Sentinel-1 imagery archived in GEE between January and March in 2015.

#### 2.2.2   Automatic weather station (AWS) observations

AWS 14, AWS 17, and AWS 18 are automatic weather stations installed and operated by the institute for marine and atmospheric research Utrecht (IMAU) and the British Antarctic survey (BAS). Shortwave radiation ($R_s$) and surface albedo ($\alpha_o$) are observed using a Kipp and Zonen CNR1 radiation sensor at AWS 14 and AWS 17, and CNR4 radiation sensor at AWS 18. The same sensor also measures down- and upwelling longwave radiation ($R_l$). Air temperature ($T_{2m}$), air pressure ($p$), and

relative humidity ($RH$) are corrected for heating of the shielded housing by solar radiation, especially in situations with low wind speed. More details on the experimental setup are found in Kuipers Munneke et al. (2018b), Smeets et al. (2018), and Jakobs et al. (2020).

#### 2.2.3   The regional climate model RACMO2

RACMO2 is a regional climate model adapted for the simulation of the weather over snow and ice surfaces, and its impact on

the surface mass and energy balance. The version used in this study is RACMO 2.3p2 (Van Wessem et al., 2018). The entire AIS is simulated at a horizontal resolution of approximately 27 x $27 \, \mathrm{km}^2$, for the period 1979–2019. For this study, we select the model output between 2009 and 2016, overlapping with the availability of MODIS and AWS observations on the Larsen Ice Shelf. RACMO2 has a scheme for calculating the evolution of surface albedo, a key parameter for the surface energy balance

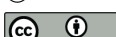



in summer, and an important factor in determining surface melt. The albedo scheme is based on the metamorphism of snow
grains determining the amount of incoming radiation that is absorbed in the snowpack. The albedo scheme does not account
for ponding meltwater, the appearance of blue ice, or other icy surfaces wind like glaze, or refrozen supraglacial water. All of
these surface types tend to have a lower albedo than a snow surface.

## 3    Methods

Essentially, we develop a deep learning model that corrects RACMO2 surface melt, based on differences in modelled and ob-
served surface albedo values. A comprehensive flowchart of the method is given in Figure 2. Input to the deep learning model
consists of relevant, predictive meteorological input and a $\Delta\alpha$ that represents the difference between observed and simulated
albedo values. The deep learning model needs to be trained (Figure 2 Block II-2), which is done on a reference data set derived
from AWS observations. To build this reference data set, the surface energy balance model is used to perturb the surface albedo
from AWS observations with an amount $\Delta\alpha$ (Figure 2 Block II-1). The model is then trained to predict the resulting change
in surface melt. The trained model is subsequently applied to RACMO2, where $\Delta\alpha$ is computed as the difference between
MODIS-observed albedo and RACMO2-simulated albedo (Figure 2 Block II-4). The MODIS albedo itself is corrected for
variations in solar zenith angle, and cloud cover (Figure 2 Block II-3) to allow comparison with the RACMO2 albedo.

The remainder of this section describes the methodology in more detail. The perturbation of the AWS observations, as de-
scribed above, is referred to as 'AWS data augmentation', a frequently-used term in deep learning. More fundamentals and
terminology in deep learning can be found in LeCun et al. (2015). In order, we present details of (1) the formulation of the
concept of the deep learning model, (2) the surface energy balance model design, (3) the augmentation (perturbation) of AWS
observations, (4) training and validation of the deep learning model, (5) preparation and correction of MODIS albedo, and (6)
the final application of the deep learning model, and its evaluation (Figure 2).

### 3.1    Concept formulation of this study

Previously, the additional absorbed solar energy that stems from the difference between a lower observed albedo and a higher
modelled albedo has been converted entirely to surface melt (Lenaerts et al., 2017). But this approach neglects the fact that all
the terms in the surface energy balance change when surface albedo is lowered. For example, it could lead to a higher surface
temperature, and thus enhanced heat loss by longwave radiation to the atmosphere. Or a decreased air temperature gradient
in the atmospheric boundary layer could diminish the amount of sensible heat added to the surface by turbulence. Therefore,
our approach makes use of (1) original, imperfect model albedo, (2) MODIS albedo observations, and (3) a full surface energy
balance model to compute the effect of a change in surface albedo on *all* energy balance terms.

Central in our approach, we assume that an imperfect RACMO2 simulation of surface melt is caused by an imperfect sim-
ulation of surface albedo in the model. Absorbed solar radiation is by far the major source of energy for surface melt in the



summertime surface energy balance of the Antarctic surface (Lenaerts et al., 2017). Surface albedo strongly modulates this amount of absorbed solar radiation. Therefore, we assume that an imperfect simulation of surface albedo is the most dominant cause of mismatches between modelled and observed surface melt (Lenaerts et al., 2017).

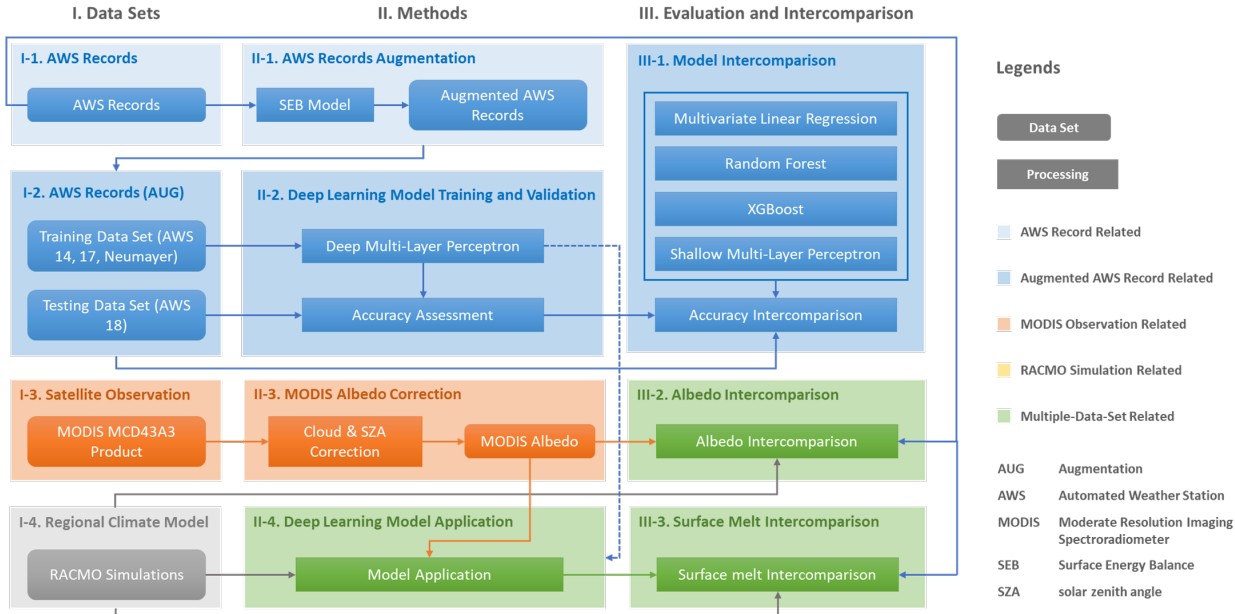

**Figure 2.** Overall flowchart illustrating the employed data sets, implemented methods, and corresponding intercomparisons and evaluations.

### 3.2 Surface energy balance model

The surface energy balance is given as

$$M_0 = S_\downarrow + S_\uparrow + L_\downarrow + L_\uparrow + H + L + G. \tag{1}$$

Here, the radiative fluxes are represented by $S_\downarrow$ and $S_\uparrow$ (incoming and reflected solar radiation), and $L_\downarrow$ and $L_\uparrow$ (downwelling and upwelling longwave radiation). $H$ and $L$ are the turbulent fluxes of sensible and latent heat, and $G$ is the ground heat flux, computed from subsurface temperature. The sum of these fluxes is 0 if the surface temperature is below the freezing point.

At the freezing point, the sum of the fluxes equals the melt energy $M_0$. All fluxes are defined positive towards the surface, and expressed in $\mathrm{W\,m^{-2}}$. The model is identical to the one described in Kuipers Munneke et al. (2012b). The model has been previously calibrated by minimizing the difference between observed and modelled surface temperature and subsurface temperatures.



### 3.3 AWS perturbation

The deep learning model needs to be trained to predict by what amount the surface melt is changed ($M_a$) if albedo is changed by an amount $\Delta\alpha$ (Figure 2, Box II-1). Therefore, we construct a reference data set by artificially raising or lowering the albedo, using the surface energy balance model to quantify how the reduced/additional amount of absorbed solar radiation results in reduced/additional melt and/or is redistributed over the other energy fluxes in the energy balance.

One of the advantages of this approach is that the perturbation strongly increases the size of the training data set, while conserving the internal consistency of the energy balance. As a positive side effect, the data augmentation increases the number of low-albedo values in the data set, as in the original AWS time-series, low-albedo values were scarce. In the flowchart in Figure 2, the data augmentation is in box II-1.

The entire time-series of AWS 14, AWS 17, and AWS 18 are adjusted by a value of $\Delta\alpha$ between -0.30 (lowering) and +0.09 (increasing) in steps of 0.03. Since positive values of $\Delta\alpha$ can result in non-physically high albedo (i.e. albedo higher than 1.0), adjusted albedos greater than 0.95 are discarded.

### 3.4 Deep learning model: the multilayer perceptron (MLP)

To estimate the change in surface melt from RACMO2, a deep MLP model is developed where we estimate the additional 160 surface melt ($M_a$) by means of regression $F$:

$$M_a = F(\alpha_s, \Delta\alpha, T_{2m}, S_\downarrow, L_\downarrow, M_0, F_m, \Delta M_t, D) \tag{2}$$

where the input parameters are the simulated albedo ($\alpha_s$) itself, the albedo difference ($\Delta\alpha \equiv \alpha_o$ - $\alpha_s$) between the observed ($\alpha_o$) and simulated albedo, air temperature at 2 m ($T_{2m}$), incoming shortwave radiation ($S_\downarrow$), downwelling longwave radiation ($L_\downarrow$), simulated surface melt ($M_0$), boolean melt flag ($F_m$), surface melt difference to the previous day ($\Delta M_t$), and record date 165 as day-of-year ($D$). As such the deep MLP model builds on all important drivers for melt fluxes. Moreover, it includes surface melt information from the previous day as memory effects can also play a role. To put emphasis on the days that the surface is actually melting, we provide an additional, boolean melt flag as input to the model.

The model is programmed in Python using *Keras*, a high-level deep learning application programming interface of *Tensor-* 170 *Flow 2.0*, where the architecture is illustrated in Figure 3. The deep MLP model consists of 15 hidden layers, each with 64 neurons. We use the adaptive moment estimation (Adam) optimizer (Kingma and Ba, 2014) with a learning rate of 0.0003. The Xavier normal initializer (Glorot and Bengio, 2010) is used to set the initial random weights of the layers. The maximum iteration is set to 5000, and an 'early stopping' is applied to avoid overfitting and monitor the variation of 'loss'. Consequently, the training process is terminated early if the 'loss' stops improving after 20 epochs. It not only improves the training efficiency 175 but also adds regularization effects to the deep MLP model. To further avoid overfitting, for all hidden layers, $L_2$ regularizations





(Ng, 2004) and dropouts (with a rate of 0.1) are applied in every third layer (Figure 3). On the other hand, to address 'gradient vanishing', three 'shortcuts' (He et al., 2016) are built to convey the residual information from previous layers. To accelerate the training process, batch processing with a batch size of 4096 is deployed using the shuffled training data set.

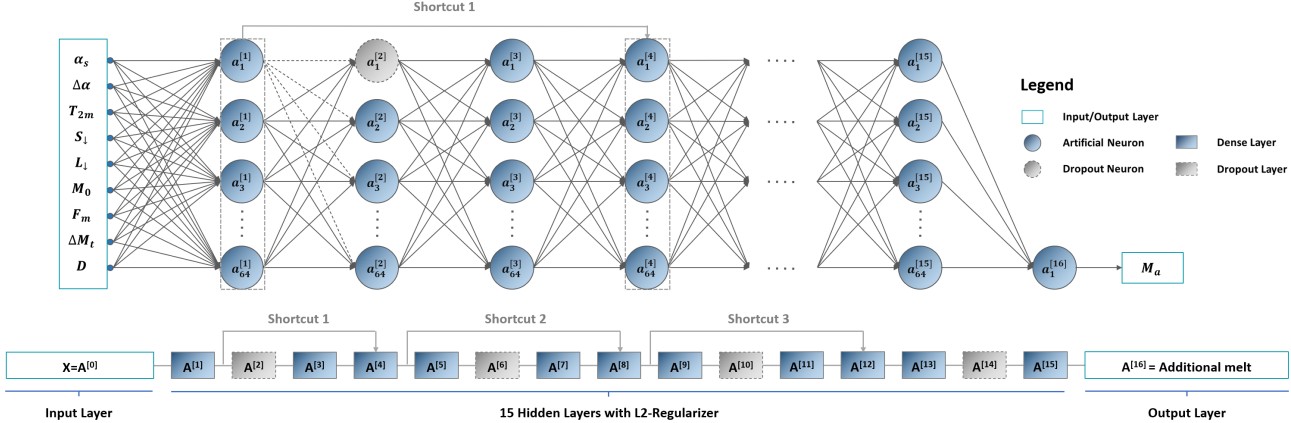

**Figure 3.** Overview of the built deep multilayer perceptron (MLP) model architecture, in which the input parameters are the simulated albedo ($\alpha_s$) itself, the albedo difference ($\Delta\alpha \equiv \alpha_o - \alpha_s$) between the observed ($\alpha_o$) and simulated albedo, air temperature at 2 m ($T_{2m}$), incoming shortwave radiation ($S_\downarrow$), downwelling longwave radiation ($L_\downarrow$), simulated surface melt ($M_0$), boolean melt flag ($F_m$), surface melt difference to the previous day ($\Delta M_t$), and record date as day-of-year ($D$).

Additionally, for a comparison purpose, we have also built a multivariate linear regression model, a boosting model (XG-Boost, Chen and Guestrin, 2016), a bagging model (random forest, Breiman, 2001), and a shallow MLP model (single hidden layer with 12 neurons). The bagging and boosting models are developed in Python using the open-source package *Sci-kit learn* and *XGBoost*, and the shallow MLP model is developed using Keras. To tune the hyper-parameters in the random forest and XGBoost model, the *GridSearchCV* function built in *Sci-kit learn* is used. Validation of the above-mentioned models is implemented by comparing the accuracy metrics based on training and validation data set. To evaluate the deep MLP model and the other machine/deep learning models, we have separated the augmented AWS observations (section 3.3) into a training data set containing AWS 14 and AWS 17, and a validation data set consisting of AWS 18. Afterward, three metrics are calculated to evaluate the model performance, i.e. root mean square error (RMSE), mean absolute error (MAE), and coefficient of determination ($R^2$).

## 3.5 MODIS albedo correction and interpolation

When the deep learning model is applied to RACMO2 output, corrections to surface melt are guided by MODIS observations of surface albedo. However, the MODIS white-sky albedo product (MCD43A3) is a clear sky product, and needs to be converted to total sky albedo by correcting for the influence of changing solar zenith angle and cloud cover, both of which have a significant impact on surface albedo over snow (Kuipers Munneke et al., 2008). Therefore, we use MODIS white sky albedo





as diffuse albedo $\alpha_s$ at 50° and applied the RACMO2 parametrisation for solar zenith correction ($d\alpha_u$) and clouds ($d\alpha_\tau$) as described in Kuipers Munneke et al. (2011) and developed in Gardner and Sharp (2010). It requires cloud optical depth and solar zenith angle as input. The latter is computed as a function of date and geographical location. Cloud optical depth ($\tau$) is calculated from RACMO2 simulations of the ice water path (IWP) and liquid water path (LWP) using the parameterization from Stephens (1978):

$$\tau_i = \frac{3}{2} \cdot \frac{IWP}{\rho_i R_{\text{eff,i}}} \tag{3}$$

$$\tau_w = \frac{3}{2} \cdot \frac{LWP}{\rho_w R_{\text{eff,w}}}, \tag{4}$$

where the subscripts $i$ and $w$ denote a separate treatment for ice and water. The total cloud optical depth is:

$$\tau = \tau_i + \tau_w, \tag{5}$$

The effective particle radius used for ice ($R_{\text{eff,i}}$) is 30 μm and for water ($R_{\text{eff,w}}$) is 13 μm (Henderson et al., 2011). The
respective densities are $\rho_i = 916.7 \ \text{kg m}^{-3}$ and $\rho_w = 1000 \ \text{kg m}^{-3}$. IWP and LWP ($\text{kg m}^{-2}$) are integrated cloud ice/water content over cloud depth $z$, under the assumption that the cloud is vertically uniform with respect to the drop-size distribution, i.e. well-mixed. Although this is a simplified assumption, it allows a first order correction for cloud effects on the satellite albedo. Once the daily total-sky MODIS albedo values were derived we implemented a linear interpolation over time to fill in missing values due to potential missing values due to persistent cloud cover.

## 3.6 Application of the MLP model to RACMO2

Since our objective is to improve surface melt simulations from RACMO2 over all Larsen ice shelves, it is vital to assess the model performance to both the reference data set, and to RACMO2 simulations directly. For the reference data set, we use the AWS data and perturbed albedo changes as input data for Eq. (2) , whereas for the RACMO2 simulations we rely on RACMO2 data and MODIS albedo observations as inputs to calculate the corrected surface melt ($M_c$):

$$M_c = max(M_0 + M_a, 0) \tag{6}$$

where $M_0$ represents uncorrected surface melt simulations for both reference and RACMO2 data sets, and $M_a$ stands for the additional surface melt estimated by the deep MLP model. Additionally, to calibrate over-corrections, we have set the negative corrected surface melt to zero using Eq. (6). Outside the austral summer when absorbed solar radiation is no longer the major source of energy for surface melt, the deep MLP model is switched off, and the original RACMO2 simulations are used to
calculate the annual surface melt.

The correction can be performed consistent with the temporal extent of the MCD43A3 product (i.e. from 16 February, 2000 to present). However, to intercompare the original RACMO2 surface melt, the corrected one, and the AWS observations, the application in this study is limited to 2009 to 2016, for which AWSs have been operated on the Larsen Ice Shelf (Figure 1).





### 3.7 Evaluation of the MLP model performance

To evaluate the deep MLP model performance, we conduct two separate analyses. The first evaluation focuses on the robustness of the method that we developed. In section 4.1, we assess the deep MLP model performance using a cross-validation, i.e. we evaluate if the deep MLP model is able to recreate the time-series of surface melt from the surface energy balance model. We set aside the reference data set at AWS 18 as the validation data set, thereby preserving the complete inter- and intra-annual melt variability. In the same way, we also benchmark the deep MLP model with other machine learning models: a multivariate linear regression model, an XGBoost model (a leading boosting machine learning model), a random forest regression (a leading bagging machine learning model), and a shallow MLP model containing only one hidden layer and few neurons (hereafter referred to as a shallow MLP model).

The second analysis, presented in section 4.3, assesses the performance of the deep MLP model in the final application to RACMO2 model simulations that are corrected using MODIS albedo observations. First, we thoroughly evaluate MODIS albedo in an intercomparison with AWS and RACMO2 albedo before we use it as an input to the deep MLP model. This can reveal potential sources of error from the input albedo in the deep MLP model application, and demonstrates discrepancies among the three data sets at different AWS locations. Second, we apply the deep MLP model to MODIS and RACMO2 data, and present the corrected surface melt in section 4.3, along with the AWS observations, QuikSCAT(QSCAT)-based estimations (Trusel et al., 2013), and original RACMO2 simulations over the Larsen Ice Shelf (AWS 14, AWS 17, and AWS 18). Additionally, to further investigate the potential influence from the imprecise meteorological input parameters to the deep MLP model, we display the contemporary $T_{2m}$, $S_\downarrow$, $L_\downarrow$, and $\alpha$ from AWS observations, RACMO2 simulations, and/or MODIS observations.

## 4 Results

### 4.1 MLP performance: Application of the MLP to AWS data

Here, we test the technical correctness of the deep MLP model, and its ability to learn the behaviour of the surface energy balance model. We first present the cross-validation results, indicating the performance of the deep MLP model when applied to data from a different location. Subsequently, we compare the performance of the deep MLP to other machine learning models. Finally, we present the capacity of the developed MLP model to reconstruct the time-series of surface melt from the surface energy balance model.

#### 4.1.1 Accuracy assessment based on cross-validation

The cross-validation of the additional daily surface melt ($M_a$) predicted by the deep MLP model (Figure 4) shows a high correlation ($R^2 = 0.95$) between the deep MLP modelled $M_a$ and that from the surface energy balance calculations based on the perturbed observations at AWS 18. The overall RMSE and MAE are 0.95 and 0.42 , respectively. However, no- and low-melt values are abundant, and they can greatly reduce the errors, especially outside of the summer season. To eliminate





such effect, we discriminate between melt and no-melt periods, and the results are summarized separately in Figure 4b and Figure 4c. During melt periods, errors are 1.05 (RMSE) and 0.70 $\mathrm{mm\,w.e.\,per\,day}$ (MAE). These values are higher than for the no-melt periods, which have an RMSE and MAE of 0.91 and 0.34 $\mathrm{mm\,w.e.\,per\,day}$, respectively. The $R^2$ is 0.21 lower in no-melt conditions because of a number of notable outliers during no-melt periods (Figure 4c). These outliers indicate that

the deep MLP model sometimes has issues simulating low values of additional melt (Figure 4a–c), leading to a 'sword-like' pattern in Figure 4a. The 'handle' of the 'sword' originates mainly from no-melt periods (Figure 4c). Such errors mostly occur when $\Delta\alpha$ is smaller than -0.2 during no-melt periods (red circles in Figure 4c), which are rare in reality. In such cases, the deep MLP model is more likely to set the surface melt back to zero. The second type of error is the horizontal line crossing the origin. It mainly occurs during melt periods in winter, between approximately May and August. The deep MLP model

does not know how to behave, because solar radiation is absent and albedo is undefined: $\Delta\alpha$ is no longer the dominant factor that influences the magnitude of the surface melt. This is confirmed by the winter melt anomalies in 2016 (see section 4.1.3). Assessment of the model performance throughout the year (Figure 4d) illustrates a bowl-like pattern for the overall accuracy and accuracy for no-melt periods, with RMSEs close to zero during the no-melt periods. During the melt season, RMSEs are close to 1.0 $\mathrm{mm\,w.e.\,per\,day}$. In Figure 4e, the model accuracy for different $\Delta\alpha$ shows a minimum for $\Delta\alpha$ equals zero. The

higher the absolute value of the $\Delta\alpha$ is, the higher the RMSE and MAE are. Lastly, when daily values of surface melt are aggregated to monthly surface melt, the accuracy of the deep MLP model increases strongly ($R^2 = 0.99$). It suggests that it is more appropriate to utilize the monthly results.

### 4.1.2 Performance of machine learning and deep learning model

The deep MLP model, in general, outperforms other machine learning models (Table 1). The performance of the multivariate
linear regression model is weakest, i.e. the highest RMSE and MAE, and the lowest $R^2$. The commonly-used random forest and XGBoost models both outperform the multivariate linear regression model, and also the shallow MLP model. Also, they are less likely to overfit after the hyper-parameter tuning and regularization. In particular, random forest regression has achieved a slightly better MAE and $R^2$ than the deep MLP model in the validation data set. In comparison to the shallow MLP architecture, the deep MLP with stricter regularization and shortcuts shows a noticeable improvement in accuracy. We thereby demonstrate

that a lot of improvement can be achieved by customizing and optimizing a deep learning model compared to taking an off-the-shelf model. It confirms that the deep neural network can improve the surface melt simulations from RACMO2 over the Larsen Ice Shelf, when there are accurate meteorological input data available.

### 4.1.3 Time-series of MLP-predicted surface melt

The time-series of corrected surface melt for different $\Delta\alpha$ show that lower values of $\Delta\alpha$ lead to higher daily surface melt
(Figure 5), except for melt events during the wintertime in 2016 (Figure 5a). Figure 5b reveals that the disparity between the deep-MLP-predicted surface melt and the augmented AWS observations increases with the drop of albedo, especially when $\Delta\alpha < -0.18$. The time-series show that most surface melt occurs during the austral summer. The largest differences between the deep-MLP-predicted surface melt and the augmented AWS observations are mainly found during peak episodes of melt. It



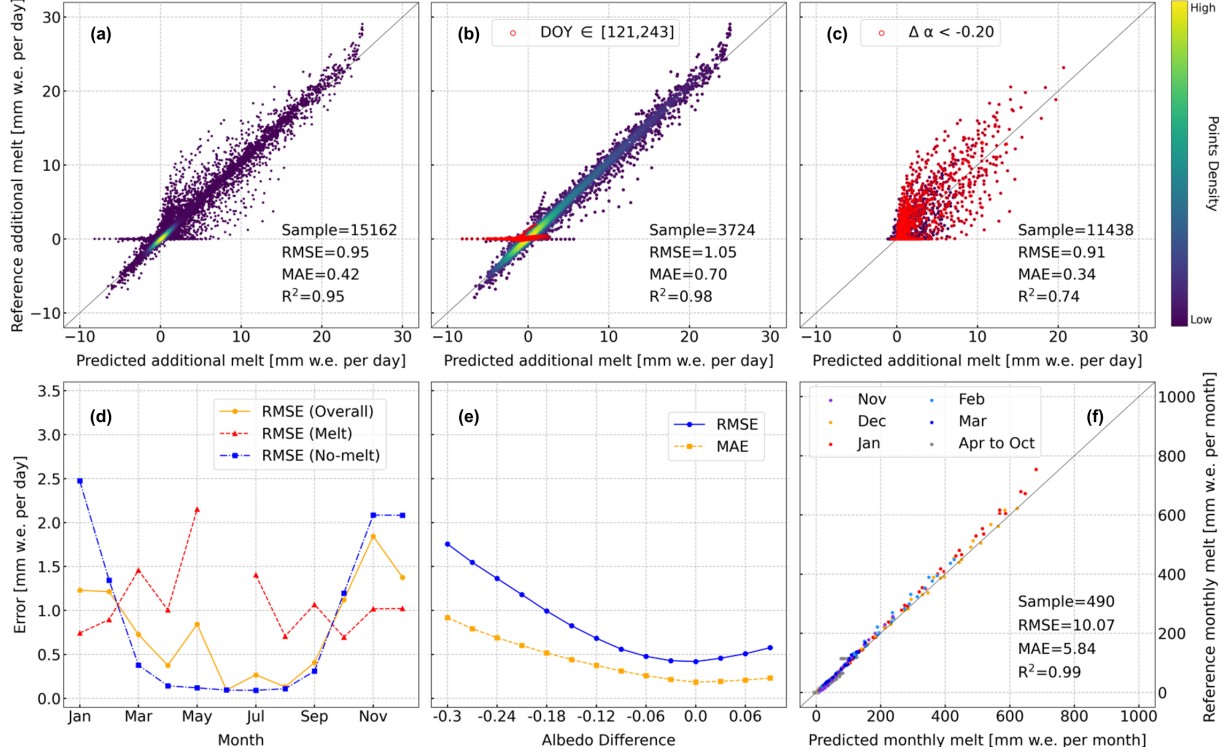

**Figure 4.** Performance of the deep multilayer perceptron (MLP) model regarding its (a) overall accuracy, (b) accuracy of melt periods, (c) accuracy of no-melt periods, (d) accuracy in each month (because no melt event occurred in June during the automatic weather station (AWS) operation period based on the AWS observations, the June value is absent), (e) accuracy under all albedo differences, and (f) monthly accuracy, in terms of the additional melt. RMSE, MAE, $R^2$, DOY, and $\Delta\alpha$ stand for root mean square error, mean absolute error, coefficient of determination, day-of-year and albedo difference, respectively.

is also noteworthy that the daily surface melt difference among different $\Delta\alpha$ values is larger than the difference between the
deep MLP results and (perturbed) AWS observations which indicates that the albedo is the main source of uncertainty, not the uncertainties from the deep MLP model.

As shown in Figure 5a, there is a period of anomalously high surface melt during the winter of 2016, demonstrated previously in AWS and satellite observations, and in RACMO2 simulations (Kuipers Munneke et al., 2018a). In the deep MLP
results, the winter melt episodes are almost identical under different $\Delta\alpha$. Given that winter melt events occurred during polar darkness, the incoming shortwave radiation is almost zero. In such circumstances, the incoming shortwave radiation is no longer the key factor leading to a surface melt increment. Therefore, $\Delta\alpha$ no longer influences such winter melt events. The actual trigger for the winter melt events is *föhn*, adiabatically heated winds that descend from the Antarctic Peninsula mountains to the west of the Larsen Ice Shelf (Marshall et al., 2006; Orr et al., 2008; Cape et al., 2015).





**Table 1.** Performance of different models in estimating additional melt of the regional atmospheric climate model version 2.3p2 (RACMO2) simulations. The training data set is the augmented automatic weather station (AWS) observations from AWS 14 and AWS 17, and the validation data set is the augmented AWS observations from AWS 18.

| Model | Training Data Set | | | Validation Data Set | | |
|---|---|---|---|---|---|---|
| | RMSE | MAE | $R^2$ | RMSE | MAE | $R^2$ |
| **Multivariate Linear Regression** | 3.04 | 2.19 | 0.51 | 3.19 | 2.35 | 0.48 |
| **XGBoost** | 0.86 | 0.38 | **0.97** | 1.00 | 0.44 | **0.96** |
| **Random Forest Regression** | 0.95 | 0.37 | 0.95 | 0.98 | **0.40** | 0.95 |
| **Shallow Multilayer Perceptron** | 1.24 | 0.65 | 0.92 | 1.32 | 0.72 | 0.91 |
| **Deep Multilayer Perceptron** | **0.77** | **0.33** | **0.97** | **0.95** | 0.42 | 0.95 |

RMSE, MAE, and $R^2$ stand for root mean square error, mean absolute error, and coefficient of determination


In Figure 5b, we demonstrate that the deep MLP model is capable of not only enhancing existing melt, but also to simulate melt when in the original time-series there was no melt. When $\Delta\alpha$ exceeds a certain value, the overall temporal melt pattern differs from that under lower $\Delta\alpha$. It results in a longer-lasting and more intense melt event. Also, because of the dependence of melt on the occurrence of melt on the previous day, the duration of a certain melt event may be prolonged. It simulates

a melt-albedo feedback: an initial melt event may trigger a melt event the next day, as albedo is reduced.

### 4.2 Comparing albedos from MODIS, AWS, and RACMO2

For clear-sky conditions (Figure 6), AWS 14 and AWS 17 show higher correlations with MODIS ($R^2$ = 0.28 and 0.20, respectively) than RACMO2, whereas for AWS 18 this is reversed with better correlations between AWS and RACMO2 ($R^2$ = 0.40). RMSE between AWS and MODIS is lower than the RMSE between AWS and RACMO2 at AWS 14 (by <0.01) and

AWS 17 (by 0.01), but 0.02 higher at AWS 18. Histograms of albedo values are shown in panels d–f in Figure 6. At AWS 14, both MODIS and RACMO2 show higher values of albedo than AWS observations. For lower albedo values, AWS observations agree better with RACMO2 simulations than with MODIS observations. Even though the three data sets have a similar range of albedo values, the MODIS albedo observations are narrow and less skewed. At AWS 17, AWS observations show a broad distribution and are mostly below 0.85. They agree better with the MODIS observations than with RACMO2 simulations. At

AWS 18, MODIS observations are lower than AWS observations and RACMO2 simulations, and RACMO2 simulations are more similar to AWS observations.

Typical time-series of albedo from RACMO2, AWS and MODIS show that the differences between the three albedo products are relatively small during most of the austral summer season (Figure 7). However, in the first half of December, MODIS

and RACMO2 observed/simulated comparably high albedo at AWS 17 (around 11 December 2013) and AWS 18 (around 6



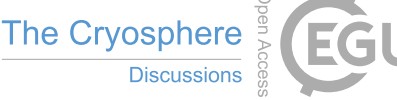

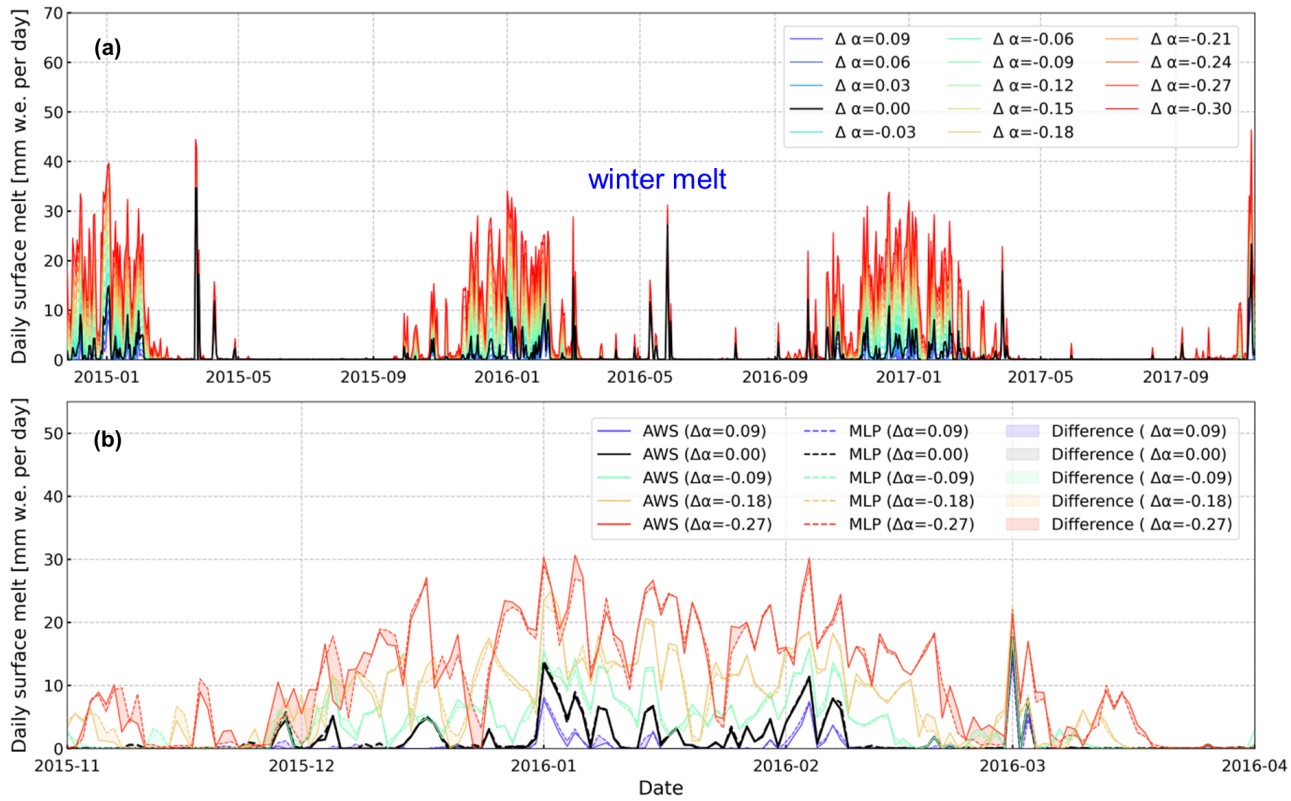

**Figure 5.** Dynamics of the corrected regional atmospheric climate model version 2.3p2 (RACMO2) surface melt using the deep multilayer perceptron (MLP) model at automatic weather station (AWS) 18, (a) under all albedo differences ($\Delta\alpha$) and (b) its subset between November 2015 and April 2016 together with the augmented AWS observations for comparison.

December 2014). The contemporary optical depth is also relatively high. Vice versa, at AWS 18 around December 11, 2014, both MODIS and RACMO2 observed/simulated comparably low albedo, and the optical depth is close to zero on a cloudy day. The difference remains low during the middle of the summer season but gradually increases towards the end of the summer season in February. RACMO2 simulations tend to produce the highest albedo at the three AWS on average. At AWS 14 and

AWS 18, RACMO2 simulations are more consistent with the AWS observations than with MODIS observations. MODIS observations are comparably lower than AWS observations and RACMO2 simulations at the end of February. On the contrary, AWS observations are much lower than RACMO2 simulations at AWS 17. For albedo values higher than 0.80, AWS observations and MODIS observations are similar, but for albedo below 0.80, AWS observations show a broader tail towards lower values. It is noteworthy that each AWS has different background geophysical settings, and the three products have very dif-

ferent spatial resolutions: AWS observations are local in-situ observations, while MODIS albedo observations and RACMO2 albedo simulations are of 27 km spatial resolution. Further analyses and discussion can be found in section 5.1.

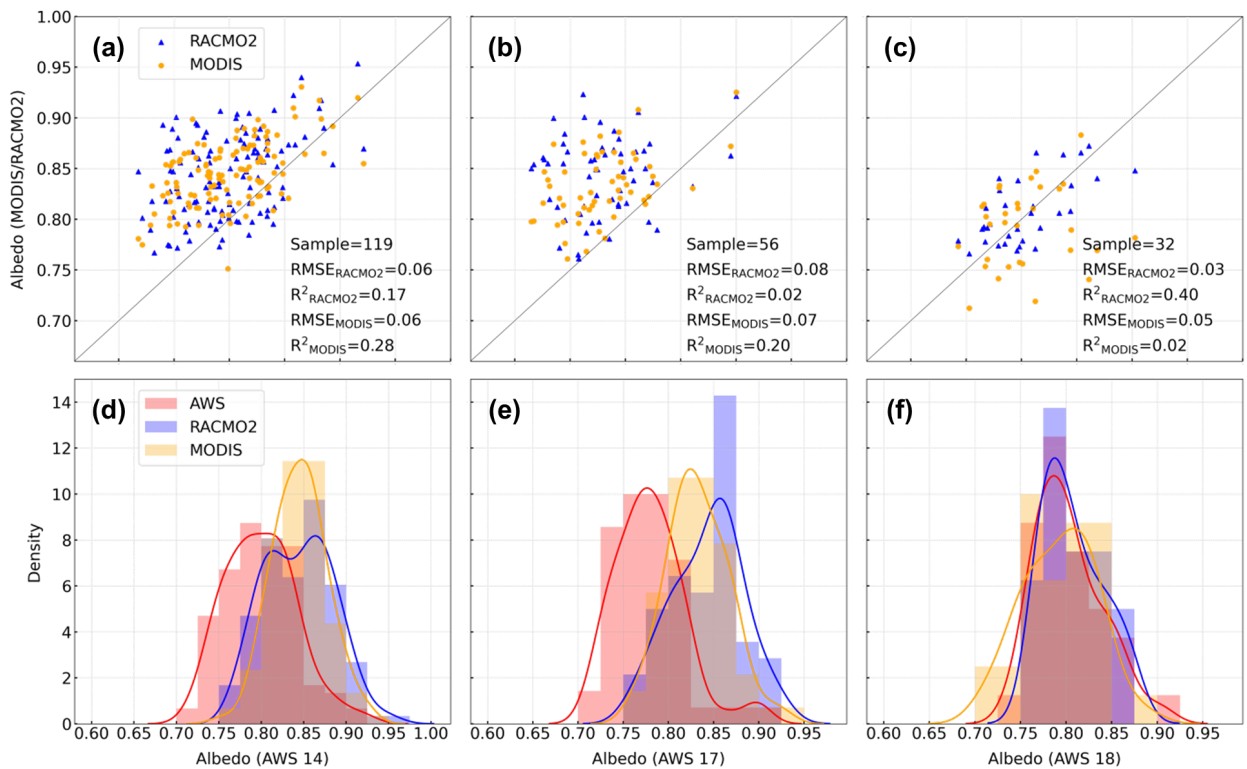

**Figure 6.** Scatter plots illustrating albedo difference under clear-sky conditions during austral summers between automatic weather station (AWS) and moderate resolution imaging spectroradiometer (MODIS) observations, as well as albedo difference between AWS observations and regional atmospheric climate model version 2.3p2 (RACMO2) simulations in (a) AWS 14, (b) AWS 17, and (c) AWS 18, together with their corresponding distributions in (d) AWS 14, (e) AWS 17, and (f) AWS 18. RMSE and $R^2$ stand for root mean square error and coefficient of determination.

To translate clear-sky MODIS albedo to an all-sky albedo, we used optical depth simulated by RACMO2 at its horizontal resolution of 27 km. The optical depth from RACMO2 and the cloudiness from the MOD09GA product are displayed in Fig-
ure 7 for AWS 14, AWS 17, and AWS 18. The optical depth from the RACMO2 simulations is often close to zero on clear-sky days, while there are more erroneously high values of optical depth observed at AWS 18 than at AWS 14 and AWS 17. It is because that RACMO2 provides daily average IWP and LWP values, however, the RACMO2.3p2 version has a systematic overestimation of both cloud ice and water, simulating clouds that are optically too thick. The general biases in these cloud variables are especially large for the coastal bins (Figure 6b in van Wessem et al. (2018)), and could thus explain the erroneously
high optical depth observed at AWS 14 and AWS 17 (Figure 7), in which clouds considered 'thin' when $\tau \leq 6$ or 'thick' for $\tau \geq 12$. The interpretation of albedo differences in this section is also hampered because values from different sources are representative of areas of very different sizes. An AWS observation has a spatial footprint of the order of 10 x 10 $\mathrm{m}^2$, whereas



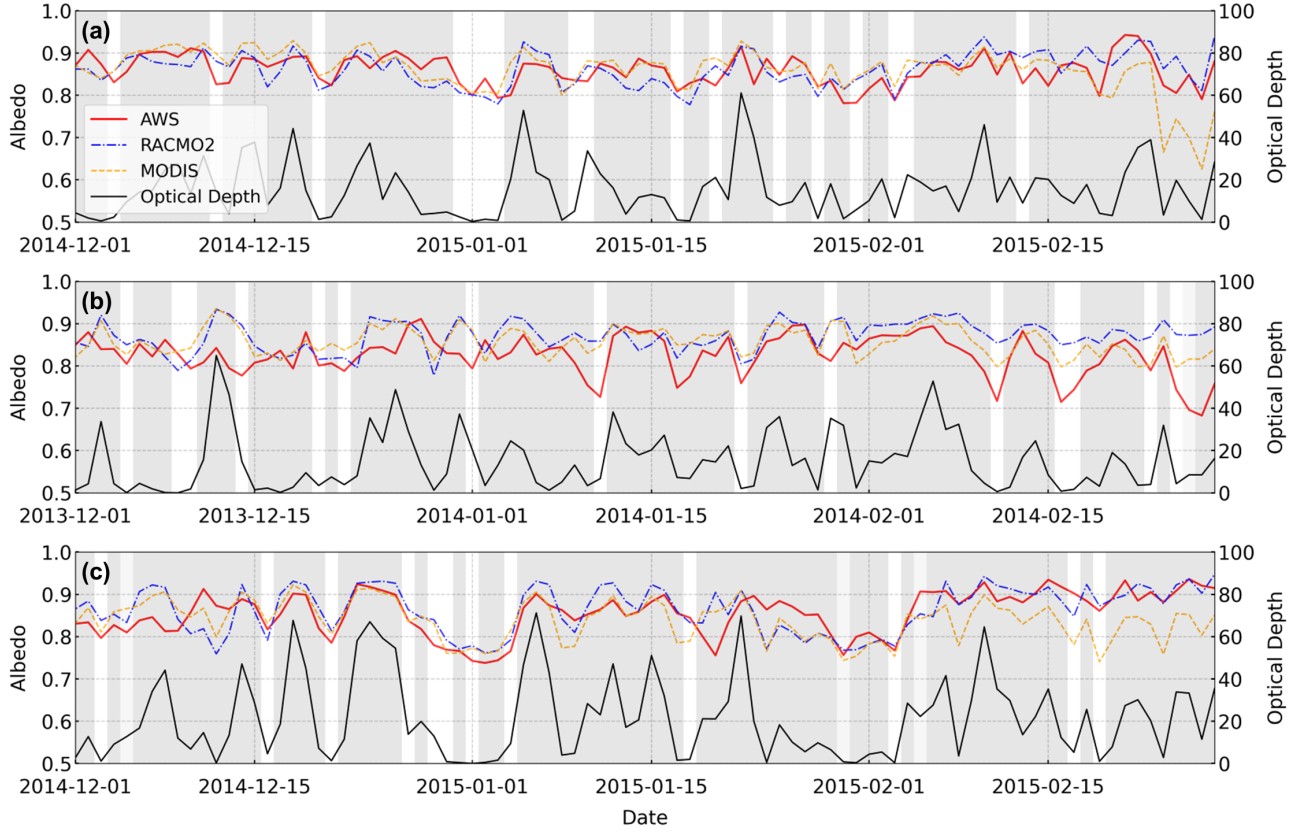

**Figure 7.** Albedo dynamics from the regional atmospheric climate model version 2.3p2 (RACMO2) simulations, automatic weather station (AWS) observations, and moderate resolution imaging spectroradiometer (MODIS) observations at AWS 14, AWS 17, and AWS 18 during the austral summer 2015/2016.

the RACMO2 and MODIS values represent an area of hundreds of squared km. We demonstrate the possible consequences for each AWS location by high-resolution maps of Sentinel-1 backscatter (Figure 8), which is produced following Luckman

et al. (2014). The orange and red pixels represent the detected melt based on $\sigma_t^\circ - \overline{\sigma_w^\circ} \lesssim -3$ dB, in which $\sigma_t^\circ$ is the backscatter value for a pixel observed at a certain date (t), and $\overline{\sigma_w^\circ}$ is the mean backscatter between June and August (Luckman et al., 2014; Trusel et al., 2012).





**Figure 8.** Spatiotemporal patterns of surface melt, represented by the drop of backscattering coefficient in Sentinel-1 imagery, in automatic weather station (AWS) 14, AWS 17, and AWS 18 between January and March in 2015, and their corresponding regional atmospheric climate model version 2.3p2 (RACMO2) pixels.

### 4.3 MLP performance: Application of the MLP to RACMO2 and MODIS data

At AWS 14, the discrepancies between the AWS observations and RACMO2 simulations in 2m air temperature and incoming shortwave radiation are the smallest amongst the three AWSs (Figure 9a). It potentially reduces the effects of these meteorological parameters in the surface melt in addition to albedo. On a daily basis (Figure 9b), the deep MLP model tries to make the melt better in agreement with observations at AWS 14, although it is difficult for the deep MLP model to alter the RACMO2 melt signal significantly. Therefore both the RACMO2 and MODIS albedo data remain similar. It is noteworthy that a melt event at the beginning of February 2014 is erroneously corrected by the deep MLP model since both AWS and RACMO2 do not observe or simulate such a melt event. The contemporary albedo simulations/observations agree well among the three data



sets. However, RACMO2 does simulate higher 2m air temperature. Therefore, we assume that the developed deep MLP model and its "learned" melt mechanism are also sensitive to the other input meteorological parameters, such as 2m air temperature. Also, the discrepancy at the daily scale seems to originate from a different timing of melt in both RACMO2 and the observations. The deep MLP model is unable to completely repair the timing offset, which is different from the observations. When considering longer-term averages, the deep MLP model delivers an improvement of the melt fluxes. Figure 9c shows a closer agreement of the deep MLP model with AWS observations for monthly melt fluxes. Moreover, for annual fluxes, both the deep MLP model and RACMO2 annual melt fluxes are higher than in the AWS observations, but both close to the QSCAT estimates.

At AWS 17, RACMO2 shows a systematic overestimation of air temperature and incoming shortwave radiation compared to AWS observations (Figure 10a). Consequently, it leads to apparent overestimations of surface melt in RACMO2 simulations in December 2013 (Figure 10b), when the discrepancy in albedo is small. During the middle of the austral summer 2013/2014, replacing the MODIS-observed albedo by the AWS-observed albedo in the deep MLP model input results in a better agreement with the AWS surface melt observations. It reveals the potential of improving the deep MLP model accuracy by better estimating the surface albedo from MODIS. At AWS 17 during the end of February, when AWS observed two extensive melt events, RACMO2 simulates the incoming shortwave radiation well. However, even by replacing the MODIS albedo observations with AWS observations in the deep MLP model input, the mismatch is still significant. It seems the underestimations in 2m air temperature is the trigger. Therefore, attention should be paid to other influencing meteorological parameters in addition to albedo. Although the deep MLP model improves the estimate of monthly surface melt (Figure 10c) and brings annual melt more in line with QSCAT (Figure 10d). But compared to the AWS observations, both the deep MLP model and RACMO2 annual melt are much lower (Figure 10d).

At AWS 18, the deep MLP model fails to improve surface melt compared to observations. Instead of reducing the overestimations of original RACMO2 simulations in the daily surface melt, the deep MLP model tends to further increase the daily surface melt (Figure 11b). It is plausibly due to the overestimations in 2m air temperature by RACMO2 simulations (Figure 11a). Besides, using AWS-observed albedo instead of MODIS-observed albedo shows an improvement in agreement with AWS surface melt. It is either because of imprecise MODIS albedo or great spatiotemporal variance between the surface melt at AWS 18 and its corresponding RACMO2 pixel. On the other hand, it indicates that the deep MLP model does not always reduce the surface melt simulations but can also increase surface melt simulations. In the end, both RMSE and MAE increase $11.32 \, \mathrm{mm \, w.e. \, per \, month}$ and $7.87 \, \mathrm{mm \, w.e. \, per \, month}$ after the correction using the deep MLP model applied to RAMCO2 and MODIS data (Figure 10c). The discrepancies are also shown in the comparison in the annual surface melt, in which the deep MLP model produces the highest estimations throughout all the years. QSCAT estimates and AWS observations are the lowest (Figure 10d).



**Figure 9.** Application of the deep multilayer perceptron (MLP) model to RACMO2 and MODIS data at automatic weather station (AWS) 14, and the results of: (a) discrepancies of input meteorological parameters among the AWS observations, regional atmospheric climate model version 2.3p2 (RACMO2) simulations, and moderate resolution imaging spectroradiomete (MODIS) observations; (b) daily surface melt time-series from the original RACMO2 simulations, AWS observations, and MLP estimations using different input data set for albedo (from either MODIS or AWS observations) and the other meteorological parameters from RACMO2 during the austral summer 2013/2014; (c) a scatter plot of monthly surface melt from the deep MLP model estimations and the original RACMO2 simulations compared to AWS observations; and (d) annual melt fluxes (July–June) at AWS 14 from observations, RACMO2, the deep MLP predictions, and QSCAT (QuikSCAT) estimations (Trusel et al., 2013).



**Figure 10.** Application of the deep multilayer perceptron (MLP) model to RACMO2 and MODIS data at automatic weather station (AWS) 17, and the results of: (a) discrepancies of input meteorological parameters among the AWS observations, regional atmospheric climate model version 2.3p2 (RACMO2) simulations, and moderate resolution imaging spectroradiomete (MODIS) observations; (b) daily surface melt time-series from the original RACMO2 simulations, AWS observations, and MLP estimations using different input data set for albedo (from either MODIS or AWS observations) and the other meteorological parameters from RACMO2 during the austral summer 2013/2014; (c) a scatter plot of monthly surface melt from the deep MLP model estimations and the original RACMO2 simulations compared to AWS observations; and (d) annual melt fluxes (July–June) at AWS 14 from observations, RACMO2, the deep MLP predictions, and QSCAT (QuikSCAT) estimations (Trusel et al., 2013).



## 5 Discussion

### 5.1 Impacts from different geophysical settings on MLP model performance

The developed MLP model shows a good performance under the ideal scenario, i.e., when only the albedo observations from an AWS are perturbed by an amount $\Delta\alpha$ (see section 3.3). Its performance has been evaluated by a cross-validation based on the reference data set at AWS 18, under different $\Delta\alpha$ for both melt and no-melt periods. Performance of the deep MLP model is good especially during the austral summer, when most of the melt events take place, caused mainly by solar radiation modulated by surface albedo. When replacing the AWS observations by MODIS observations and RACMO2 simulations, the deep MLP performance is more difficult to establish (section 4.3) and varies for different AWS locations. The first complicating factor is our assumption that albedo is the main driver for surface melt differences. Although we assumed that $\Delta\alpha$ is the key factor in surface melt variation during the austral summer in Antarctica, in fact, surface melt is also determined by other meteorological parameters (e.g., air temperature), which can be biased in RACMO2 (section 4.3). As a second complicating factor, the performance of the deep MLP model is compromised because of systematic biases in MODIS albedo compared to AWS observations. It seems that the geophysical setting (geolocation, surface type, melt pattern, topographical characteristics) play an important role in both of these factors. Therefore, we discuss the geophysical settings of AWS 14, AWS 17, and AWS 18 in order to explain the deep MLP results.

**Scenario 1 (AWS 14):** AWS 14 and its corresponding RACMO2 pixel are located central in the northern part of the Larsen C Ice Shelf, where the terrain is flat and homogeneous, and covered by snow and firn. At AWS 14, albedo is relatively well simulated by RACMO2, and the discrepancies among the three albedo data set are low throughout the austral summer (Figure 7). No constant over-/underestimations in albedo has been found in MODIS observations and RACMO2 simulations, compared to AWS observations. Owing to the homogeneous geophysical settings, its spatiotemporal melt pattern is homogeneous as well. Therefore, when applying the deep MLP model to RACMO2 and MODIS, the result shows a better agreement to the AWS observations at AWS 14 than at AWS 17 and AWS 18.

**Scenario 2 (AWS 17):** AWS 17 is located in the middle of Scar Inlet, a remainder of the Larsen B Ice Shelf, where the terrain is flat. However, its corresponding RACMO2 pixel also contains the grounded areas at the edge of the pixel, and partially covers mountainous terrain. The albedo discrepancy is the largest amongst the three data set (i.e. AWS observations, MODIS observations, and RACMO2 simulations). At the point level, AWS 17 is located in the most extensive melt area, resulting in AWS observations that are constantly lower than both MODIS observations and RACMO2 simulations for a 27 km pixel. On the other hand, albedo from RACMO2 simulations and MODIS observations remains close, since the proportion of the grounding line and mountains is low in the pixel. Therefore, the deep MLP model results are less in agreement than those at AWS 14.

**Scenario 3 (AWS 18):** AWS 18 is situated in an inlet near the grounding line of the the Larsen C Ice Shelf, the surrounding area



consists of complex terrain, and also contains grounded areas. Its corresponding RACMO2 pixel is also the most mixed pixel
among the three AWS locations, and melt ponds can occur during melt events. The AWS at this location does record extensive
melt events, but it is not located in the area of strongest melt in the RACMO2 pixel. The discrepancies of albedo from the
three data sets are small at the beginning and the middle of austral summer, but MODIS observes much lower albedo at the
end of austral summer. The spatiotemporal melt pattern is also the most heterogeneous among the three AWS locations. Apart
from the $\Delta\alpha$, the terrain itself also impacts the surface melt process. In the end, the challenges in accurately deriving albedo at
AWS 18 and the combination of multiple influencing factors result in poor performance of the deep MLP model. Furthermore,
attention must be paid to the representativeness of the AWS observations in such an area, which makes the verification difficult.

**Scenario 4 (additional site north of AWS 17):** The last scenario is an additional location north of AWS 17 on Scar Inlet,
sitting on the grounding line of the Larsen B Ice Shelf (Figure 12). Such a location presents an ideal scenario where the actual
albedo is systematically lower than RACMO2 simulations (Figure 12). This is a situation that the deep MLP model is actually
designed for. As a result, the daily surface melt corrected by the deep MLP model is also systematically higher than the original
RACMO2 simulation. It is an ideal area to illustrate the expected performance of the deep MLP model where the overestima-
tion in surface albedo is the dominant factor causing surface melt underestimation in RACMO2. However, we do lack AWS
observations to evaluate MLP performance at this location.

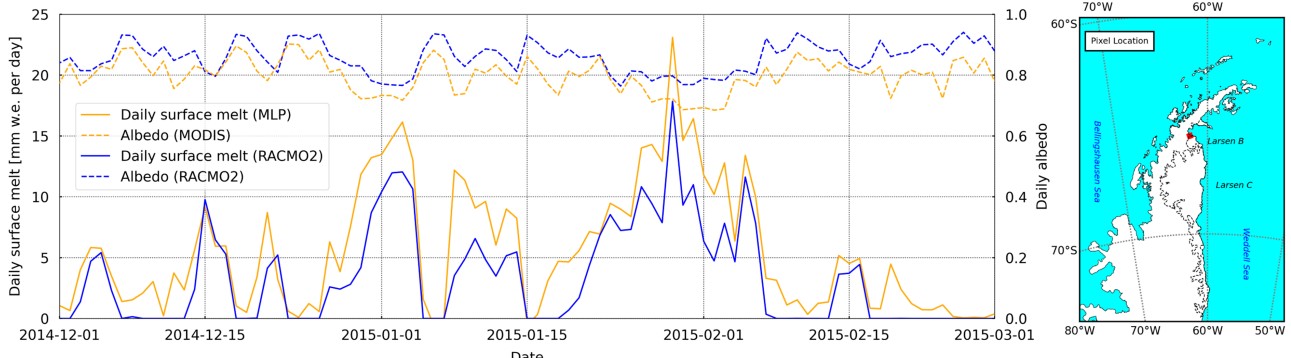

**Figure 12.** Daily surface melt from regional atmospheric climate model version 2.3p2 (RACMO2) simulations and multilayer perceptron
(MLP) estimations, and albedo from RACMO2 and MODIS at the additional test site in the north of AWS 17 in the left panel, and its
geolocation (i.e. the red square in the right panel).

## 5.2 Lessons learned from this study over the Larsen Ice Shelf

Our study shows that the deep MLP model has the potential to improve surface melt estimates in Antarctica. On the other
hand, the results at AWS 18 shed light on that complicated and heterogeneous terrains can result in poor performance of the
deep MLP model. AWS 18 is a typical area has high-melt features that we are interested in, but such high-melt features are
usually of small-scale. Although AWS observations suggest there is overestimation in albedo simulations from RACMO2, it is



challenging for MODIS to correct it when implemented in a coarse resolution climate model. Therefore, we need to combine the low-resolution RACMO2 with higher-resolution MODIS to resolve this issue. Apart from this, there is also a potential to further increase the resolving power of our method, including: (1) a further refinement of the deep-learning-based framework, (2) improvement of the albedo derivation from satellite observations, both in magnitude and spatial resolution, and (3) examination of MLP model performance over (blue) ice surfaces.


Refinement of the deep-learning-based framework includes developing a module to switch the deep MLP correction, and testing the state-of-the-art deep learning architectures and models. Given that the concept of correcting surface melt in non-albedo-driven areas may reduce the agreement between the AWS observations and RACMO2 simulations, it is necessary to develop an application strategy switching the deep MLP correction in different areas. The strategy can take topographical

characteristics (e.g., elevation, aspect, slope), albedo difference, geolocation into consideration, since these factors can be influential (section 5.1). It is also worth mentioning that the deep learning model implemented in this study is fundamental in order to prove the concept. Exploring other state-of-the-art deep learning architectures and models, such as the recurrent neural network, can be beneficial for accuracy improvement.

Improvement of the albedo derivation from satellite observations can result in better corrected surface melt results using the deep MLP model. Attention should be paid to the generation and correction of the MODIS albedo observations, as the MCD43A3 is a 16-day synthetic product. For a certain day, a composite 500 m resolution daily product is generated based on 16-day records centered on the given day (Schaaf and Wang, 2015). Temporal interpolation over long periods can lead to high disparity in the albedo during an ablation season, especially on high-melt days. Moreover, in heterogeneous terrain such as the

location of AWS 18, albedo may be highly different in space. Therefore, implementing a spatial resampling may improve this issue. Also, the MODIS albedo correction for cloud cover based on optical depth from RACMO2 needs additional care, since in some locations RACMO2 overestimates optical depth.

On the Larsen Ice Shelf, reduction in albedo is mainly due to the aging of snow and firn. Yet, in icy areas (mainly on the

ice shelves of eastern Antarctica and the Transantarctic Mountains), albedo reduction is much stronger than over firn and snow surfaces. Moreover, the surface energy balance over ice surface can be fundamentally different from that over firn and snow surfaces for which the deep MLP model is trained in this study. Therefore, the deep MLP model performance in blue ice areas needs to be examined with extra care.

In addition to the improvement of the deep MLP model, we also need to look for novel and unconventional ways to validate results over unsurveyed areas. Earth observation satellites, such as Landsat, MODIS, advanced very-high-resolution radiometer (AVHRR), can provide temporal record of the land surface and its modification over the last decades at different spatial and temporal resolutions. Potentially, we can compare the surface melt outcomes to proxies from remotely sensed data, e.g., melting decibel days (Trusel et al., 2013), lake depth (Philpot, 1989) and volume (Moussavi et al., 2020).



## 6 Conclusions

This paper demonstrates that surface melt simulations from the regional climate model RACMO2 can be improved by deploying the deep learning model trained on the automatic weather station (AWS) observations. The deep learning model takes meteorological parameters, day-of-year, and the original RACMO2 surface melt simulations as the inputs. The cross-validation shows a good performance (RMSE = 0.95 mm w.e. per day, MAE = 0.42 mm w.e. per day, and $R^2$ = 0.95) in the Larsen Ice Shelf based on the augmented AWS observations. Regarding accuracy, the deep learning model outperforms some leading machine learning models (random forest regression and XGBoost) and a shallow deep learning model. To address the problem regarding inaccurate albedo simulations from RACMO2 model, MODIS albedo observations after cloudiness and solar zenith angle correction have been used. The corrected MODIS albedo observations show a better correlation with AWS observations than the RACMO2 simulations at AWS 14 and AWS 17. At AWS 18, large disparities have been identified between corrected MODIS albedo observations and AWS observations. Possible explanations are the local geophysical settings (including cloudiness, spatiotemporal melt pattern, and topographical characteristics) and imperfect albedo correction at AWS 18. Finally, to correct the surface melt simulations from RACMO2 over the entire Larsen Ice Shelf, assessing model performance in areas and periods without AWS observations is indispensable. For deep learning model application, meteorological parameters except for albedo from AWS observations are replaced by the RACMO2 simulations. The corrected MODIS albedo observations replace albedo observations from AWS. The results indicate that the deep learning model performs well in the area with a homogeneous spatiotemporal melt pattern (AWS 14) and in the area with a heterogeneous spatial melt pattern which is yet homogeneous through time (AWS 17). The model performance is of high uncertainty in the areas with a heterogeneous spatial melt pattern which varies through time (AWS 18) due to the inaccurate albedo input and the lack of ground truth data. In summary, the concept of correcting surface melt simulations from the regional climate model RACMO2 using deep learning is feasible. Future studies are still required to refine the deep-learning-based framework, improve the albedo derivation from satellite observations, examine the deep MLP model performance over the (blue) ice surface, and develop a novel validation scheme.

*Author contributions.* ZH, PKM and SL designed the study. PKM implemented the AWS perturbation, ZH, SL, and MI corrected the MODIS albedo, and ZH designed and trained the deep neural network architecture. ZH, PKM, SL, and MB carried out quantitative analyses. PKM, SL, and MB discussed the deep neural network results. ZH and PKM wrote the manuscript with contributions from all co-authors.

*Competing interests.* The authors declare that they have no conflict of interest.

*Acknowledgements.* The authors would like to thank Dr. Luke D. Trusel for providing monthly surface melt estimations derived from QuikSCAT. This research was funded by the Netherlands Space Office (NSO) grant ALWGO.2018.039. PKM is funded by the Nether-



lands Earth System Science Center (NESSC). MI was supported by Dutch Research Council (NWO) / Netherlands Space Office Grant

ALWGO.2018.043, whereas SL was funded by.





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
