# Peer review of "Improving Surface Melt Estimation over Antarctic Ice Sheet Using Deep Learning: A Proof-of-Concept over the Larsen Ice Shelf"

_The Cryosphere, 2021_

## Referee Comment (RC2)

Zhongyang Hu et al. 2021: Improving surface melt estimates over Antarctica Using Deep Learning.

In this article, authors develop a deep machine learning model with the aim of improving surface melt estimates from a regional climate model, RACMO2. The ML model is applied to three locations on the Larsen C (and remnants of Larsen B) ice shelf, which adds complexity to the scientific question of whether the ML model can improve surface melt estimates. The technique is novel, and likely to lead to further investigation of ML techniques for improving model simulations of various glaciological mechanisms. The study is well thought out and evaluated, and the results are presented clearly. Although I list numerous specific concerns below, they are quite minor, and I would be happy to see the publication of this manuscript once they are implemented.

Specific comments

Ln 22: I would include additional references for hydrofracture where they explain this process more thoroughly than the IMBIE paper. Perhaps Kuipers Munneke et al. 2014 (https://doi.org/10.3189/2014JoG13J183) or Gilbert and Kittel 2021 (https://doi.org/10.1029/2020GL091733).
Ln 26: here you talk about melt volume importance, but you haven't mentioned melt volumes yet. You have only talked about melting in terms of mass loss and ice shelf stability. I would perhaps open this paragraph with an estimate of sea level rise associated with AIS melt volumes (either present estimates or future projections).
Ln 34: change to 'face difficulties in accurately estimating surface melt…'
Ln 41+44: change to 'physically-based models'
Ln 58: In number 2 you contradict why Larsen C is ideal. Rather than say that high-quality observations are scarce in Antarctica, instead say that high-quality observations are available in this region, which is rare for Antarctica.
Ln 64: remove first 'in'
Ln 67: some contradiction between 'no ice shelves' on line 65 and 'almost only' on line 67. The wording also needs changing as 'almost only' isn't right. Perhaps 'Ice shelves on the Antarctic Peninsula are mostly located on the eastern coast.
Ln 70-73: You need some citations here for these values.
Ln 82: indicator **of** surface melt
Ln 94: wording needs consideration. How can a model be adapted for its impact on SMB and SEB?  The model doesn't have an impact on the SMB. Perhaps you mean adapted for more accurate representation of SMB?
Ln 95: Was RACMO2 forced by ERA-Interim or ERA5? Do you have a figure of the domain that you used for Larsen C?
Ln 100-103: Any citation for the albedo scheme so that readers can investigate further?
Ln 106: change to 'and the difference between observed and simulated albedo values ($\Delta\alpha$)'
Ln 209: How often has this interpolation had to occur due to persistent cloud cover? How many days of missing values are there? How frequent is the MODIS overpath at this location and at what approximate time of day? How does the correction to MODIS vary during the winter with a much lower solar zenith angle for a persistent time?

Ln 212: I would specify that you mean the MLP model here- as RACMO2 is also a model, so the current sentence 'it is vital to assess the model performance' could be misunderstood as referring to RACMO performance.

Ln 217: Can additional surface melt only be positive? As in your introduction you mentioned that it was also important to correct RACMO where it overestimates melt. Is this why you set negative corrected melt to 0, so that melt as a whole cannot be negative, or only the change in melt cannot be negative. Perhaps this needs rephrasing.

Ln 218: Did you attempt to apply the model outside of the austral summer? Did you turn on the model specifically at December 1 and off again at February 28/29? What about in years where the melt season starts early or ends late, which can be the case (e.g 2010), especially in the presence of föhn winds: see King et al. 2017. (https://doi.org/10.1002/2017JD026809). What difference could the model have outside of the summer?

Ln 264: Earlier you say that the MLP is not applied outside of austral summer, yet here you discuss May and August. So is the MLP applied year round? Or are the winter values from RACMO without being corrected?

Ln 308: Include the R2 for RACMO too, so that the reader than read that correlations are higher.

Ln 320-331: Point to some figures or tables here, or include some results of statistics to back up your analysis, as it is currently quite qualitative.

Ln 357: Are you able to say why it was erroneously corrected?

Ln 361: Have others also found timing offsets between RACMO and observations previously? A citation would strengthen this section. Perhaps this is covered more in the discussion though.

Ln 371: In which year?

Ln 380-390: AWS18 is located in a region with blue ice, where albedo is generally low and the valleys are relatively narrow. It could be that RACMO2 fails to capture the blue ice zone and could also have land use discrepancies with the topography poorly resolved in 27km resolution. It could be useful here to mention the blue ice zone and/or include some references to this, as it would be unlikely if RACMO2 was able to represent these surface conditions.

Ln 435: 'actual albedo' is a little misleading as the reader is unsure whether this comes from AWS observations or MODIS. You have to read the Figure caption to understand. I would write (from MODIS) or something similar in the text.

Figure 7: What are the white and grey bars in the background? Include this info in the caption

---

## Author Comment (AC1)

**Response to comments from Reviewer 1**

**General comments:**

This paper demonstrates that the estimations of the Antarctic ice sheet surface melt volumes by the regional climate model RACMO2 can be improved significantly if a deep multilayer perceptron (MLP) model is employed to correct the RACMO2 simulation results. In recent years, various types of machine learning techniques, which include the MLP model, also known as artificial intelligence (AI) are widely used in many directions in accordance with improved performance of computers as well as rapid increases in relevant data size. Therefore, the topic addressed by the authors is opportune. As far as I know, such an attempt is very new in the glaciological research community. In this respect, this study is of high value. In addition, this informative paper is well written and organized. Therefore, this reviewer recommends its publication in the journal TC although several points in the paper can be improved prior to its publication. Please find below my specific comments and suggestions on the paper.

We thank the reviewer for taking valuable time to review this manuscript, and for providing very thoughtful and constructive comments for improving its quality. We carefully considered all of the reviewer's suggestions and modified the manuscript accordingly. For a better overview and clarity we numbered and colored the reviewer's comments, as follows:

(0) The comments from the reviewer are colored in black.
Our answer to each comment is colored below in blue.
The proposed changes to the original manuscript are then highlighted in red.

We hope that the reviewer agrees that we took her or his suggestions into serious consideration, and that we adapted the manuscript in a satisfying way.

**Specific comments:**

(1) Title: Suggest rephrasing "Antarctica" → "Antarctica Ice Sheet". I think it is better to state explicitly that "ice sheet" surface melt is investigated in this study.

We agree that it would be better to narrow down the tittle to 'Antarctica Ice Sheet'.

Improving Surface Melt Estimation over  ***Antarctic Ice Sheet*** Using Deep Learning: A Proof-of-Concept over the Larsen Ice Shelf

(2) L. 3: "simulations": simulations with a climate model? Please clarify.

Yes, the term 'simulations' in Line 3 is confusing, we have clarified it as suggested.

This study aims to demonstrate the potential of improving surface melt simulations ***with a regional climate model*** by deploying a deep learning model.

(3) L. 11: "XGBoost", Its definition is needed here. In my opinion, most readers of the journal TC are not so familiar with machine learning methods.

Thanks for pointing out this issue. We agree that such a terminology can confuse readers who are not familiar with machine learning, and it might be too detailed for abstract. Thus, we decide to remove it from the abstract, but in the main text, we have added a short definition in addition to its reference in Line 180-181.

Moreover, the deep MLP model outperforms conventional machine learning models  and a shallow MLP model.

L. 180 ... we have also built a multivariate linear regression model, a boosting model (XGBoost  *: a highly effective and widely-used tree boosting system;* Chen and Guestrin, 2016), ...

(4) L. 14: "the heterogeneous drivers of melt": Its intention is unclear to me. Can the authors describe it more concretely?

The term 'heterogeneous drivers' refers to air temperature, topography, katabatic wind, to name a few; they have been added to the text.

However, for one location (AWS 18) the deep MLP model does not show improved agreement with AWS observations, likely  *because surface melt is driven to a large extent by other factors (e.g., air temperature, topography, katabatic wind) than albedo alone*.

(5) L. 45: "remote sensing albedo observations" It is worth to mention MODIS here.

We agree with the reviewer. It has been added in the text, as suggested.

..., and remote sensing albedo observations *from the moderate resolution imaging spectroradiometer (MODIS)*.

(6) L. 60: "scatterometry": Can the authors specify the type of the sensor used for the comparison?

Yes, we agree with the reviewer. We have included the full name of the scattermetry, i.e., QuikSCAT, in the text.

... and (3) a previous comparison between RACMO2 albedo and  *radar backscatter from the Quick Scatterometer (QuikSCAT)* revealed that ...

(7) L. 60 ∼ 61: "RACMO2 melt can be both higher and lower than observations": depending on what conditions? More detailed explanations are useful for readers.

Yes we agree with the reviewer. A comprehensive comparison between the QuikSCAT scatterometry and RACMO2 is in the paper by Trusel et al. (2013). We merely want to highlight that RACMO2 is not not only biased in one direction. In this area, we see RACMO2 both overestimating and underestimating surface melt. Therefore, it is a good test for the application of our model that should be able to both enhance and reduce melt estimates from RACMO2.

... revealed that **_both positive and negative values of the difference between RACMO2 and observed surface melt exist in this area_** (Trusel et al., 2013).

(8) Figure 1 caption: "The elevation information ": "The elevation information for tundra areas "?

The digital elevation model shown in Fig. 1 does not only provides information about mountainous areas but also the lowlands, however, these ice surfaces are very homogeneous, resulting in mono-colored (cyan) areas. We have changed the figure caption as below:

 **_Surface elevation is_** derived from the ETOPO1 1 arc-minute global relief model ...

(9) L. 85: Section 2.2.2: I would like to confirm whether the authors considered possible factors that can disturb albedo measurements (e.g., tilt of the AWS mast and/or sensors, surface hoar, etc) and corrected the measured albedos.

The solar radiation observations are corrected for sensor tilt and riming. In Line 91, we changed the text to:

More details of the experimental setup **_and data corrections_** are found in Kuipers Munneke et al. (2018b), Smeets et al. (2018), and Jakobs et al. (2020).

(10) L. 85: Please explain whether the ice surface appears or not during the austral summer ablation period at each AWS.

To our knowledge, no bare ice areas appear at the surface at these AWS locations.

(11) L. 89 ~ 90: "Air temperature (T2m), air pressure (p), and relative humidity (RH) ": It is better to indicate that they are measured at the surface.

Yes, we agree. We have clarified it in our text.

The same sensor also measures down- and upwelling longwave radiation ($R_l$). Air temperature ($T_{2m}$), air pressure (p), and 90 relative humidity (RH) **_measured at 1-4 m above the surface_** are corrected for heating of the shielded housing by solar radiation, especially in situations with low wind speed.

(12) L. 98: "evolution of surface albedo": The following description on "surface albedo" (The albedo scheme is based on the metamorphism of snow grains determining the amount of incoming radiation that is absorbed in the snowpack) is only on "snow albedo". Therefore, indicating explicitly like "evolution of snow albedo" is better.

The reviewer is correct. We have changed the text as suggested.

RACMO2 has a scheme for calculating the evolution of  **_snow_** albedo, a key parameter for the surface energy balance in summer, and an important factor in determining surface melt.

(13) L. 111 ∼ 112: "The MODIS albedo itself is corrected for variations in solar zenith angle, and cloud cover (Figure 2 Block II-3) to allow comparison with the RACMO2 albedo.": Its intention is a bit unclear. In the previous part, the authors state that white-sky albedo from MODIS is used in this study, so, can I assume that the authors converted white-sky albedo to (blue-sky) albedo in this process? Please clarify.

The reviewer's assumption is correct, and we agree that it is unclear here. Previously, clarification was given at a later stage in Section 3.5. Now, it has been also briefly included here.

The MODIS *white-sky* albedo is *converted to blue-sky albedo by*  *correcting*  variations in solar zenith angle, and cloud cover (Figure 2 Block II-3) to allow comparison with the RACMO2 *blue-sky* albedo which is also a blue-sky product.

(14) Figure 2 caption: What is the difference between solid and dashed lines?

The dashed line is the line that intersects more than three other lines. To avoid misunderstanding, we have now changed the dashed line into a solid one, for consistency.

[Figure]

Figure 1: Old Figure 2 on the left, and new Figure 2 on the right.

(15) L. 141 ∼ 143: It is useful for readers if the authors briefly describe how the calibration was performed.

The least constrained settings in the surface energy model is the roughness length for turbulent exchange of energy. It is varied until the bias between observed and modelled temperatures is minimized.

The model  *settings (mainly for turbulent exchange of energy) were calibrated* by minimizing the difference between observed and modelled surface temperature and subsurface temperatures.

(16) L. 161: As seen later in Fig. 7, discrepancies between in-situ measured and MODIS albedos are often found at all the AWSs. I would like to know the authors' thoughts regarding whether the deep MLP model performance can be improved if the accuracy of MODIS albedo (e.g., RMSE) is somehow considered here (e.g., input monthly RMSE values of MODIS albedo). If the authors can add some comments on this issue in Sect. 5, it will be useful for readers.

The reviewer is absolutely correct. That is also the reason why we have performed correction in two different modes (i.e., MLP: MODIS-observed albedo + RACMO2 and MLP: AWS-observed albedo + RACMO2) in Section 4.3. In Fig. 9b, Fig. 10b, and Fig. 11b, they show that once we replace the MODIS albedo by AWS albedo, the corrected surface melt is much closer to the observed ones. These findings are summarized in Section 4.3.

(17) L. 192: "the MODIS white-sky albedo product (MCD43A3) is a clear sky product": In my humble opinion, this description is incorrect: White-sky albedo is defined as albedo in the absence of a direct component when the diffuse component is isotropic.

We agree on the definition of the white-sky albedo. Here, we are referring to the fact that the MODIS MCD43A3 product is based on the Level 2 Surface Reflectance Product (MOD09, MYD09), which is a cloud-cleared (i.e. clear sky) and atmospherically corrected product. In this way the MODIS (white-sky) albedo product is a clear sky product that cannot be measured in cloudy conditions. We have clarified in the text.

 **Therefore, we use** the MODIS  albedo product (MCD43A3)**, which** is a clear sky product **based on observed clear sky surface reflectances from the Level 2 Surface Reflectance Product (MOD09, MYD09).**  **The MODIS white-sky albedo hence** needs to be converted to  **blue**-sky albedo by correcting for the influence of changing solar zenith angle and cloud cover, both of which have a significant impact on surface albedo over snow (Kuipers Munneke et al., 2008).

(18) L. 258: "0.21 lower in no-melt conditions": compared to what? It seems to me that the authors compare melt and no-melt periods here. If so, the value must be 0.24.

Yes, the reviewer is correct. We were comparing the 'no-melt' ($R^2$=0.74) to the overall condition ($R^2$=0.95), but it would make more sense to compare no-melt to melt conditions. We have changed the corresponding text in the manuscript.

The $R^2$ is  **0.24** lower in no-melt conditions than **melt conditions** because of a number of notable outliers during no-melt periods (Figure 4c).

(19) L. 261: "'handle' of the 'sword'": I could understand the intention of "sword"; however, the intention of "handle" is unclear to me.

We admit that the term 'handle' can lead to confusion. Thus, we have replaced it by a more detailed explanation, i.e., points along/close to x and y axes, in the text.

The  **points along/close to x and y axes** of the 'sword' originates mainly from no-melt periods (Figure 4c).

(20) L. 281: "neural network": This technical name appears for the first time in this paper here. I think it should be introduced much earlier in this paper.

Yes, we agree. We have add a definition of deep learning in the introduction section.

L. 38 ... Therefore, we propose a deep learning method that uses the albedo observations from remote sensing to correct for the shortcomings of climate models. **Deep learning is a machine learning technique extracting multiple levels**

*of abstraction/representation of data based on multiple processing layers, i.e., artificial neural networks (LeCun et al., 2015).* To date, deep learning has been widely applied in Earth system science to analyze and correct mismatches between model simulations and observations (Reichstein et al., 2019) as they execute much faster than physics-based models.

(21) L. 307: "clear-sky": How did the authors confirm the "clear-sky conditions"? Also, do the authors mean this is in-situ measured information? Or model simulation results? Please clarify.

The clear-sky conditions are evaluated based on the daily cloud mask provided by MODIS MOD09GA product. We have clarified it in the text.

In L. 79 ... to obtain observational information about cloud coverage at AWS locations, cloud classifications are taken from the MOD09GA daily surface reflectance product (i.e. 'MODIS/Terra Surface Reflectance Daily L2G Global 1 km and 500 m SIN Grid', Vermote and Wolfe, 2015), also archived in GEE.

For clear-sky conditions *indicated by MOD09GA* (Figure 6), AWS 14 and AWS 17 show higher correlations with MODIS ($R^2 = 0.28$ and 0.20, respectively) than RACMO2, whereas for AWS 18 this is reversed with better correlations between AWS and RACMO2 ($R^2 = 0.40$).

(22) L. 333: clear-sky" → "white-sky"; "all-sky" → "blue-sky". Also see my comment on L. 192.

We agree to use "white-sky" and "blue-sky" here.

To translate  *the white-sky* MODIS albedo to  *the blue-sky* albedo, we used optical depth simulated by RACMO2 at its horizontal resolution of 27 km.

(23) L. 381  382: "It is plausibly due to the overestimations in 2m air temperature by RACMO2 simulations (Figure 11a).": If the deep MLP model is implemented in RACMO2, and a fully two-way coupled configuration between RACMO2 and the deep MLP model becomes possible, do the authors think that the overestimation can be solved? Please discuss.

It would be very unlikely that the MLP model would be two-way coupled to RACMO2. Theoretically, the best solution would be to nudge the simulation of surface albedo in RACMO2 to a time series of MODIS albedo observations. This would be the best two-way coupling between model and observations. However, in a technical sense, this is a very problematic setup. And, from another perspective, surface albedo is an important parameter that should be allowed to evolve due to the snow physics in the model, rather than due to an observational constraint.

(24) L. 452: "the state-of-the-art deep learning architectures and models": Please specify them more in detail.

We have explained "the state-of-the-art deep learning architectures and models", as below:

Refinement of the deep-learning-based framework includes developing a module to switch the deep MLP correction, and testing the state-of-the-art deep learning architectures and models*, e.g., applying recurrent neural network architectures such as long short-term memory (LSTM; Hochreiter and Schmidhube, 1997) and transformer (Vaswani*

*et al., 2017) to help the deep learning model take in more temporal information, and/or using a convolutional neural network to learn a better representation of MODIS albedo information within a RACMO2 grid.*

Hochreiter, S. and Schmidhuber, J., 1997. Long short-term memory. Neural computation, 9(8), pp.1735-1780.

Vaswani, A., Shazeer, N., Parmar, N., Uszkoreit, J., Jones, L., Gomez, A.N., Kaiser, Ł. and Polosukhin, I., 2017. Attention is all you need. In Advances in neural information processing systems (pp. 5998-6008).

**Technical corrections:**

When refereeing a figure in running text, the abbreviation "Fig." should be used when it appears in running text and should be followed by a number unless it comes at the beginning of a sentence. See the following link for more in detail: https://www.the-cryosphere.net/submission.html#figurestables Please check throughout the manuscript carefully again.

Thanks very much for this detailed information about formatting. We will systematically change the 'Figure' to 'Fig.' when citing a figure in the running text.

L. 12: "the resulting,": The comma is unnecessary?

Yes, the comma can be deleted. Now, it has been removed.

When applying the trained deep MLP model over the entire Larsen Ice Shelf, the resulting, corrected RACMO2 surface melt shows ...

L. 63: "The Antarctic Peninsula is the mildest region of Antarctica " → "The Antarctic Peninsula (Fig. 1) is the mildest region of Antarctica "

Changed as suggested.

The Antarctic Peninsula *(Fig. 1)* is the mildest region of Antarctica, ...

L. 81: "demonstrate": I think using "confirm" instead of "demonstrate" is better here.

Changed as suggested.

To  *confirm* the spatiotemporal melt pattern in the study area, ...

L. 82: "as an indicator surface melt" → "as an indicator of surface melt"

Sorry for the grammar error, we have added the missing 'of' as suggested.

..., we derived the backscattering coefficient drops as an indicator *of* surface melt ...

L. 86: "AWS 14, AWS 17, and AWS 18 are automatic weather stations " → "AWS 14, AWS 17, and AWS 18 (Fig. 1) are automatic weather stations "

Changed as suggested.

AWS 14, AWS 17, and AWS 18 **_(Fig. 1)_** are automatic weather stations installed and operated by ...

L. 87: "Shortwave radiation" Add "down- and upwelling" before this.

We agree it is necessary to explain the type of shortwave radiation. We think it might be better to term it as: incoming/reflected shortwave radiation.

British Antarctic survey (BAS).  **_Incoming/reflected shortwave_** radiation ( $S_\downarrow$ **_and_** $S_\uparrow$), and ...

L. 107: "done" → "performed"; Using a formal word is better in a scientific paper.

Changed as suggested.

The deep learning model needs to be trained ( **_Fig. 2_** Block II-2), which is  **_performed_** on a reference data set derived from AWS observations.

L. 138: "the ground heat flux" → "the subsurface conductive heat flux"; I assume the authors do not consider interaction between the ice sheet and the ground (bedrock).

We think it might be better to term it as 'the ground heat flux at the snow surface'.

..., and G is the ground heat flux **_at the snow surface_**, ...

L. 252: "Figure 4" → "Fig. 4a"

Changed as suggested.

The cross-validation of the additional daily surface melt (Ma) predicted by the deep MLP model ( **_Fig. 4a_**) shows ...

L. 254: "0.95 and 0.42" Indicate these values with units in the text.

The missing units have been added.

The overall RMSE and MAE are 0.95 **_mm w.e. per day_** and 0.42 **_mm w.e. per day_**, respectively.

Figure 5 caption: In my humble opinion, "Dynamics of" should be rephrased to something like "Temporal changes in"

Changed as suggested.

**Figure 5.**  **_Temporal changes in_** the corrected regional atmospheric climate model version 2.3p2 (RACMO2) surface melt ...

Figure 7 caption: Same as the comment on the Figure 5 caption.

Changed as suggested.

**Figure 7.** **_Temporal changes in_** albedo  from the regional atmospheric climate model version 2.3p2 (RACMO2) simulations, ...

Figures 10 and 11: Most of the captions can be omitted by referring to the caption of Fig. 9.

The captions of Fig. 10-11 has been modified as suggested.

Figure 9. Application of the deep multilayer perceptron (MLP) model to RACMO2 and MODIS data at automatic weather station (AWS) 14, and the results of: (a) discrepancies of input meteorological parameters among the AWS observations, regional atmospheric climate model version 2.3p2 (RACMO2) simulations, and moderate resolution imaging spectroradiometer (MODIS) observations; (b) daily surface melt time-series from the original RACMO2 simulations, AWS observations, and MLP estimations using different input data set for albedo (from either MODIS or AWS observations) and the other meteorological parameters from RACMO2 during the austral summer 2013/2014; (c) a scatter plot of monthly surface melt from the deep MLP model estimations and the original RACMO2 simulations compared to AWS observations; and (d) annual melt fluxes (July–June) at AWS 14 from observations, RACMO2, the deep MLP predictions, and QSCAT (QuikSCAT) estimations (Trusel et al., 2013).

Figure 10. As Figure 9, but for AWS 17.

Figure 11. As Figure 9, but for AWS 18

---

## Author Comment (AC2)

**General comments:**

In this article, authors develop a deep machine learning model with the aim of improving surface melt estimates from a regional climate model, RACMO2. The ML model is applied to three locations on the Larsen C (and remnants of Larsen B) ice shelf, which adds complexity to the scientific question of whether the ML model can improve surface melt estimates. The technique is novel, and likely to lead to further investigation of ML techniques for improving model simulations of various glaciological mechanisms. The study is well thought out and evaluated, and the results are presented clearly. Although I list numerous specific concerns below, they are quite minor, and I would be happy to see the publication of this manuscript once they are implemented.

We thank the reviewer for the acknowledgement of the value of our work, the valuable time and attention dedicated to the manuscript, and the constructive remarks that will help us to improve the manuscript. We modified the manuscript accordingly and put the reviewers' suggestions into practice. For a better overview and clarity we numbered and colored the reviewer's comments, as follows:

(0) The comments from the reviewer are colored in black.Our answer to each comment is colored below in blue.The proposed changes to the original manuscript are then highlighted in red.

We hope that the reviewer finds the suggestions taken into serious consideration and converted into elaborative changes in the manuscript.

**Specific comments:**

(1) Ln 22: I would include additional references for hydrofracture where they explain this process more thoroughly than the IMBIE paper. Perhaps Kuipers Munneke et al. 2014 (https://doi.org/10.3189/2014JoG13J183) or Gilbert and Kittel 2021 (https://doi.org/10.1029/2020GL091733).

Thanks for the supplementary references. We agree that these additional references would better support of the statement in relation to hydrofracture, hence they have been added:

At present, mass loss is driven mainly by ice shelf weakening due to basal melt (The IMBIE team et al., 2018) or damage processes (Lhermitte et al., 2020) or hydrofracturing due to surface melt (*Gilbert and Kittel 2021; Kuipers Munneke et al. 2014;* The IMBIE team et al., 2018).

(2) Ln 26: here you talk about melt volume importance, but you haven't mentioned melt volumes yet. You have only talked about melting in terms of mass loss and ice shelf stability. I would perhaps open this paragraph with an estimate of sea level rise associated with AIS melt volumes (either present estimates or future projections).

We have added the estimate of sea level rise associated with AIS melt volumes (future projections) from the IPCC AR6 report.

...increasing the incidence of surface melt-related instability of ice shelves. In the coming centuries, surface melt is projected to increase strongly over Antarctica, increasing the incidence of surface melt-related instability of ice shelves. The *intergovernmental panel on climate change (IPCC) estimated the contribution from Antarctic Ice Sheet loss* mass to global mean sea level rise until 2100 in its recent sixth assessment report (IPCC AR6; Fox-Kemper et al., 2021). Under different shared socioeconomic pathways (SSPs), the contribution will likely be 0.03-0.27 m (SSP 1-2.6), 0.03-0.29 m (SSP 2-4.5), and 0.03-0.34 m (SSP 5-8.5) (Fox-Kemper et al., 2021). In this context, accurate information about surface melt can directly enhance our understanding of the AIS evolution and its contribution to sea level rise.

Fox-Kemper, B., H. T. Hewitt, C. Xiao, G. Aalgeirsdóttir, S. S. Drijfhout, T. L. Edwards, N. R. Golledge, 10 M. Hemer, R. E. Kopp, G. Krinner, A. Mix, D. Notz, S. Nowicki, I. S. Nurhati, L. Ruiz, J-B. Sallée, A. B. A. 11 Slangen, Y. Yu, 2021, Ocean, Cryosphere and Sea Level Change. In: Climate Change 2021: The Physical 12 Science Basis. Contribution of Working Group I to the Sixth Assessment Report of the Intergovernmental 13 Panel on Climate Change [Masson-Delmotte, V., P. Zhai, A. Pirani, S. L. Connors, C. Péan, S. Berger, N. 14 Caud, Y. Chen, L. Goldfarb, M. I. Gomis, M. Huang, K. Leitzell, E. Lonnoy, J.B.R. Matthews, T. K. 15 Maycock, T. Waterfield, O. Yelekçi, R. Yu and B. Zhou (eds.)]. Cambridge University Press. In Press.

(3) Ln 34: change to 'face difficulties in accurately estimating surface melt...'

Correction has been implemented, as suggested.

(Regional) climate models, on the other hand, face facing difficulties to in accurately estimate estimating surface melt over areas with low surface albedo.

(4) Ln 41+44: change to 'physically-based models'

Correction has been implemented, as suggested.

To date, deep learning has been widely applied in Earth system science to analyze and correct mismatches between model simulations and observations (Reichstein et al., 2019) as they execute much faster than physics-based physically-based models.

Our study aims to develop a novel framework correcting the model-observation mismatch of surface melt in the AIS with a deep learning model, which utilizes inputs from the physics-based physically-based model, ...

(5) Ln 58: In number 2 you contradict why Larsen C is ideal. Rather than say that high-quality observations are scarce in Antarctica, instead say that high-quality observations are available in this region, which is rare for Antarctica.

We have rephrased the sentence as suggested.

location for developing a framework to improve surface melt estimates, because (1) there is abundant melt; (2) highquality multi-year AWS data suitable for melt calculations (i.e. including the surface radiation budget) are scarce in *available over the Larsen Ice Shelf*, Antarctica (Jakobs et al., 2020) and ... (6) Ln 64: remove first 'in'

We apologize for this careless typo. The redundant 'in' has been removed.

... into the Southern Ocean. in In the western part of the Antarctic Peninsula the atmospheric circulation is northwestsoutheast, ...

(7) Ln 67: some contradiction between 'no ice shelves' on line 65 and 'almost only' on line 67. The wording also needs changing as 'almost only' isn't right. Perhaps 'Ice shelves on the Antarctic Peninsula are mostly located on the eastern coast.

We admit that these statements are ill-considered, we have changed it as recommended.

In the western part of the Antarctic Peninsula the atmospheric circulation is northwest-southeast, leading to mild conditions, no *few* ice shelves and little sea ice. Conversely, in the eastern Antarctic Peninsula, the circulation is south-north, resulting in colder conditions, extensive ice shelves and year-round sea ice cover. Therefore, ice shelves extend from the Antarctic Peninsula almost only at its eastern coast *ice shelves on the Antarctic Peninsula are mostly located on the eastern coast*.

(8) Ln 70-73: You need some citations here for these values.

We have added the citations as required, and the references for the corresponding values are listed, as blow:

Luckman, A., A. Elvidge, D. Jansen, B. Kulessa, P. Kuipers Munneke, J. King and N. E. Barrand. 2014. Surface melt and ponding of Larsen C Ice Shelf and the impact of foehn winds. Antarct. Sci., 26(6), 625-635

Trusel, L. D., Frey, K. E., Das, S. B., Kuipers Munneke, P., and Van Den Broeke, M. R.: Satellite-based estimates of Antarctic surface meltwater fluxes, Geophysical Research Letters, 40, 6148–6153, https://doi.org/10.1002/2013GL058138, 2013.

Turton, J.V., Kirchgaessner, A., Ross, A.N., King, J.C. and Kuipers Munneke, P., 2020. The influence of föhn winds on annual and seasonal surface melt on the Larsen C Ice Shelf, Antarctica. The Cryosphere, 14(11), pp.4165-4180.

On average, the annual melt 70 exceeds 400 mm w.e. (*Trusel et al., 2013; Turton et al., 2020*) in these inlets, distributed over about 100 melt days (*Luckman et al., 2014*). But also further east on the Antarctic Peninsula ice shelves, surface melt rates are high compared to most other ice shelves in Antarctica, at 200 to 300 mm w.e. per year (*Trusel et al., 2013*).

(9) Ln 82: indicator of surface melt

Sorry for the grammar error, we have added the missing 'of' as suggested.

To demonstrate the spatiotemporal melt pattern in the study area, we derived the backscattering coefficient drops as an indicator  $\underline{of}$  surface melt ...

(10) Ln 94: wording needs consideration. How can a model be adapted for its impact on SMB and SEB? The model doesn't have an impact on the SMB. Perhaps you mean adapted for more accurate representation of SMB?

The reviewer is correct. We have corrected the statement, as below:

RACMO2 is a regional climate model adapted for the simulation of the weather over snow and ice surfaces, and its impact on the for a more accurate representation of surface mass and energy balance.

(11) Ln 95: Was RACMO2 forced by ERA-Interim or ERA5? Do you have a figure of the domain that you used for Larsen C?

The applied RACMO2 is forced by ERA-Interim. We have added it in the text, to clarify it. The domain that was used for Larsen C are all the RACMO2 pixels covering the Larsen Ice Shelf. For a validation and intercomparison purpose, only the pixels corresponds to AWS 14 and AWS 18 are used. They are illustrated in the zoom-in in Fig. 1.

The version used in this study is RACMO 2.3p2 forced by ERA-Interim (Van Wessem et al., 2018).

(12) Ln 100-103: Any citation for the albedo scheme so that readers can investigate further?

The required reference has been added:

Kuipers Munneke, P., van den Broeke, M. R., Lenaerts, J. T. M., Flanner, M. G., Gardner, A. S., and van de Berg, W. J.: A new albedo parameterization for use in climate models over the Antarctic ice sheet, J. Geophys. Res. (D), 116, https://doi.org/10.1029/2010JD015113, 2011.

The albedo scheme does not account for ponding meltwater, the appearance of blue ice, or other icy surfaces wind like glaze, or refrozen supraglacial water. All of these surface types tend to have a lower albedo than a snow surface *(Kuipers Munneke et al., 2011)*.

(13) Ln 106: change to 'and the difference between observed and simulated albedo values  $(\Delta \alpha)$ '

We have changed the order of the words as recommended to improve the readability.

Input to the deep learning model consists of relevant, predictive meteorological input and a  $\Delta \alpha$  that represents the difference between observed and simulated albedo values the difference between observed and simulated albedo values values ( $\Delta \alpha$ ).

(14) Ln 209: How often has this interpolation had to occur due to persistent cloud cover? How many days of missing values are there? How frequent is the MODIS overpath at this location and at what approximate time of day? How does the correction to MODIS vary during the winter with a much lower solar zenith angle for a persistent time?

(14.1) How often has this interpolation had to occur due to persistent cloud cover? How many days of missing values are there?

The frequency of the interpolation applied to the prepossessed MCD43A3 data set between the austral summer 2000/01

and 2015/16 at the three AWSs are: 15.51% (AWS 14), 8.45% (AWS 17), and 1.32% (AWS 18), respectively. These are equivalent to 224 days (AWS 14), 122 days (AWS 17), and 19 days (AWS 18) of total days of missing values. These statistics have been added in the text.

**(14.2) How frequent is the MODIS overpath at this location and at what approximate time of day?**

The frequency of the MODIS overpath at a same location is twice a day: Terra crosses the equator from north to south at roughly 10:30 a.m. local time. Aqua crosses the equator from south to north at roughly 1:30 p.m. local time. (https://nsidc.org/data/modis/index.html, assessed on 12 October, 2021).

**(14.3) How does the correction to MODIS vary during the winter with a much lower solar zenith angle for a persistent time?**

During the winter, we did not implement the MLP-correction due to the absence of MODIS albedo data. To deal with the lower solar zenith angle period prior and posterior to the austral summer time, the MLP-correction is also not performed.

Once the daily total-sky MODIS albedo values were derived we implemented a linear interpolation over time (with a frequency of 15.51% for AWS 14, 8.45% for AWS 17, and 1.32% for AWS 18) to fill in missing values due to potential missing values due to persistent cloud cover.

(15) Ln 212: I would specify that you mean the MLP model here- as RACMO2 is also a model, so the current sentence 'it is vital to assess the model performance' could be misunderstood as referring to RACMO performance.

We have clarified this sentence as suggested.

Since our objective is to improve surface melt simulations from RACMO2 over all Larsen ice shelves, it is vital to assess the *MLP* model performance to both the reference data set, and to RACMO2 simulations directly.

(16) Ln 217: Can additional surface melt only be positive? As in your introduction you mentioned that it was also important to correct RACMO where it overestimates melt. Is this why you set negative corrected melt to 0, so that melt as a whole cannot be negative, or only the change in melt cannot be negative. Perhaps this needs rephrasing.

Additional surface melt can be negative (Fig. 4), but the (corrected) surface melt cannot be negative. According to Eq. 6, if the corrected surface melt (i.e., sum of  $M_0$  and  $M_a$ ) is negative, then the surface melt is set back to 0. We have rephrased it, to avoid misunderstanding.

Eq. 6:  $M_c = max(M_0 + M_a, 0)$

... Additionally, to calibrate over-corrections, *if the corrected surface melt is negative*, we have set the negative corrected surface melt *it* to zero using Eq. (6).

(17) Ln 218: Did you attempt to apply the model outside of the austral summer? Did you turn on the model specifically at December 1 and off again at February 28/29? What about in years where the melt season starts early or ends late, which can be the case (e.g 2010), especially in the presence of föhn winds: see King et al. 2017. (https://doi.org/10.1002/2017JD026809). What difference could the model have outside of the summer?

Unfortunately, we are not able to apply the model outside of the austral summer, because there is not observation of albedo during winter. Regarding the period in between (e.g., November and March), the albedo are observed under very low solar zenith angle by MODIS, which leads to in accurate albedo observations. Therefore, it also not ideal to apply the model. During these periods, the original RAMCO2 simulations will be used.

During training, we did train the MLP model to learn the surface melt process outside austral summer, since the albedo are manually tuned in the training data set. We have discovered that the model was able to learn the winter melt suggested by the reviewer. And we hope it can be considered (i.e., 'transfer-learned') by the model within the austral summer.

(18) Ln 264: Earlier you say that the MLP is not applied outside of austral summer, yet here you discuss May and August. So is the MLP applied year round? Or are the winter values from RACMO without being corrected?

It is because we have two parts of MLP model development, i.e., training/validation (Block II-2 in Fig. 2), and application (Block II-4 in Fig. 2). During the MLP training/validation, we want the MLP model to learn as much as the melt mechanism as possible, e.g., winter melt due to foehn and its relation to albedo (Fig.5). It can be implemented since the albedo is tuned manually during MLP training/validation. Hence we do have input observed albedo ( $\alpha_0$ ) outside the austral summer. While during the application, MODIS-observed albedo is required as the input. Therefore, it is impossible to apply the model outside the austral summer (in winter and the time when solar zenith angle is low). Line 264 belongs to validation, so MLP is applied.

(19) Ln 308: Include the  $R^2$  for RACMO too, so that the reader than read that correlations are higher.

We have included the required  $R^2$  in the text.

For clear-sky conditions (Figure 6 Fig. 6), AWS 14 and AWS 17 show higher correlations with MODIS ( $R^2 = 0.28$  and 0.20, respectively) than RACMO2 ( $R^2 = 0.17$  and 0.02, respectively), ...

(20) Ln 320-331: Point to some figures or tables here, or include some results of statistics to back up your analysis, as it is currently quite qualitative.

We agree that it would be better to present our result in a quantitative way there. Details regarding numbers and figures are now included in the Line 320-331, as below.

Typical time-series of albedo from RACMO2, AWS and MODIS show that the differences between the three albedo products are relatively small during most of the austral summer season (Figure 7 Fig. 7). The RMSEs between AWS-observed and MODIS-observed/RACMO2-simulated albedo in December and January are around 3.5% (AWS 14), 5.5% (AWS 17), and 4.5%, respectively. The RMSE between AWS-observed and RACMO2-simulated albedo increases up to 8.8% (at AWS 17 between RACMO2 simulations and AWS observations) in February. However At a daily basis, in the first half of December, MODIS and RACMO2 observed/simulated comparably high albedo at AWS 17 (4.5% (RACMO2) and 8.5% (MODIS) higher than AWS observations around on 11 December 2013 in Fig. 7b) and AWS 18 (6.8% (RACMO2) and 3.5% (MODIS) higher than AWS observations around on 6 December 2014 in Fig. 7c). The contemporary optical depth is also relatively high (15.64 at AWS 17, and 16.76 at AWS 18). Vice versa, at AWS 18 around on 12 December 11 2014, both MODIS and RACMO2 observed/simulated comparably low (11.4% (RACMO2) and 7.4% (MODIS) lower than AWS observations) albedo, and the optical

depth is close to zero on a cloudy day. The difference remains low during the middle of the summer season but gradually increases (up to 19.18% between RACMO2 and AWS observed at AWS 17 on 27 February 2014) towards the end of the summer season in February. RACMO2 simulations tend to produce the highest albedo at the three AWS on average. At AWS 14 and AWS 18, RACMO2 simulations are more consistent with the AWS observations than with MODIS observations (*Fig. 7a and 7c*). MODIS observations are comparably lower than AWS observations and RACMO2 simulations at the end of February. On the contrary, AWS observations are much lower than RACMO2 simulations at AWS 17 (*Fig. 7b*). For albedo values higher than 0.80, AWS observations and MODIS observations are similar, but for albedo below 0.80, AWS observations show a broader tail towards lower values (*e.g., shown in Fig. 7a in the end of February 2014 at AWS 14*). It is noteworthy that each AWS has different background geophysical settings, and the three products have very different spatial resolutions: AWS observations are local in-situ observations, while MODIS albedo observations and RACMO2 albedo simulations are of 27 km spatial resolution. Further analyses and discussion can be found in section 5.1.

(21) Ln 357: Are you able to say why it was erroneously corrected?

It is plausibly due to the significant higher temperature simulated by RACMO2 than the one observed by AWS (Fig. 9a). We have added the explanation to the text.

It is noteworthy that a melt event at the beginning of February 2014 is erroneously corrected by the deep MLP model since both AWS and RACMO2 do not observe or simulate such a melt event. *It is plausibly due to the significant higher temperature simulated by RACMO2 than the one observed by AWS (Fig. 9a).*

(22) Ln 361: Have others also found timing offsets between RACMO and observations previously? A citation would strengthen this section. Perhaps this is covered more in the discussion though.

We are not aware of others discussing this issue specifically, but RACMO2 is forced at its lateral boundaries by ERA-Interim data, and its atmosphere is allowed to evolve freely. It can lead to timing differences of weather systems of up to 1-2 days.

(23) Ln 371: In which year?

It is in year 2014, we have added it in the text.

At AWS 17 during the end of February *in 2014*, when AWS observed two extensive melt events, RACMO2 simulates the incoming shortwave radiation well.

(24) Ln 380-390: AWS18 is located in a region with blue ice, where albedo is generally low and the valleys are relatively narrow. It could be that RACMO2 fails to capture the blue ice zone and could also have land use discrepancies with the topography poorly resolved in 27km resolution. It could be useful here to mention the blue ice zone and/or include some references to this, as it would be unlikely if RACMO2 was able to represent these surface conditions.

We have also considered potential influences from blue ice or low-albedo surface, therefore, we mentioned the necessity of examining MLP performance in blue ice areas in the future in Section 5.2. To our knowledge, there is no blue ice

at AWS 18, but only some melting ponds occurred occasionally nearby. We have double checked it in the Quantarctica (Matsuoka et al., 2021) blue ice product classified using the method proposed by Hui et al. (2014). The product also suggests no blue ice occurrence at AWS 18, or only a small portion of blue ice remote from AWS 18 as suggested by the reviewer.

Moreover, this discrepancy indicates the importance of high spatial resolution corrections. At the 27 km resolution, the MODIS albedo is often lower than the RACMO2 albedo resulting in increased melt. At the point scale of the AWS, the AWS albedo is higher than the RACMO2 albedo resulting in decreased melt. This indicates that the spatial scale of corrections (27 km versus local scale) matters.

Hui, F., Ci, T., Cheng, X., Scambo, T.A., Liu, Y., Zhang, Y., Chi, Z., Huang, H., Wang, X., Wang, F., et al., 2014.526Mapping blue-ice areas in antarctica using etm+ and modis data. Annals of Glaciology 55, 129–137.

Matsuoka, K., Skoglund, A., Roth, G., de Pomereu, J., Griffiths, H., Headland, R., Herried, B., Katsumata, K., Le Brocq,536A., Licht, K., et al., 2021. Quantarctica, an integrated mapping environment for antarctica, the southern ocean, and537sub-antarctic islands. Environmental Modelling Software 140, 105015.

The discrepancies are also shown in the comparison in the annual surface melt, in which the deep MLP model produces the highest estimations throughout all the years. QSCAT estimates and AWS observations are the lowest (Figure 10d). *Moreover, this discrepancy indicates the importance of high spatial resolution corrections. At the 27 km resolution, the MODIS albedo is often lower than the RACMO2 albedo resulting in increased melt. At the point scale of the AWS, the AWS albedo is higher than the RACMO2 albedo resulting in decreased melt. This indicates that the spatial scale of corrections (27 km versus local scale) matters.*

(25) Ln 435: 'actual albedo' is a little misleading as the reader is unsure whether this comes from AWS observations or MODIS. You have to read the Figure caption to understand. I would write (from MODIS) or something similar in the text.

We agree the term 'actual albedo' is not suitable, we have replaced it by 'albedo observed by MODIS' as suggested.

Such a location presents an ideal scenario where the actual albedo *observed by MODIS* is systematically lower than RACMO2 simulations (Figure 12) (Fig. 12).

(26) Figure 7: What are the white and grey bars in the background? Include this info in the caption

The grey bars indicates the binary cloudiness of a 500  $\times$  500 MODIS pixels corresponding to each AWS location. This information has been added to the corresponding figure caption.

Figure 7. Albedo dynamics from the regional atmospheric climate model version 2.3p2 (RACMO2) simulations, automatic weather station (AWS) observations, and moderate resolution imaging spectroradiometer (MODIS) observations at AWS 14, AWS 17, and AWS 18 during the austral summer 2015/2016. The grey bars indicates the binary cloudiness of a 500  $\times$  500 MODIS pixels corresponding to each AWS location.